# Solvent-pair surfactants enabled assembly of clusters and copolymers towards programmed mesoporous metal oxides

Wenhe Xie [1,2], Yuan Ren[1], Fengluan Jiang [1], Xin-Yu Huang [1], Bingjie Yu[1], Jianhong Liu[1], Jichun Li[1], Keyu Chen [1], Yidong Zou[1], Bingwen Hu [3] & Yonghui Deng [1,2] ✉

Organic-inorganic molecular assembly has led to numerous nano/mesostructured materials with fantastic properties, but it is dependent on and limited to the direct interaction between host organic structure-directing molecules and guest inorganic species. Here, we report a "solvent-pair surfactants" enabled assembly (SPEA) method to achieve a general synthesis of mesostructured materials requiring no direct host-guest interaction. Taking the synthesis of mesoporous metal oxides as an example, the dimethylformamide/water solvent pairs behave as surfactants and induce the formation of mesostructured polyoxometalates/copolymers nanocomposites, which can be converted into metal oxides. This SPEA method enables the synthesis of functional ordered mesoporous metal oxides with different pore sizes, structures, compositions and tailored pore-wall microenvironments that are difficult to access via conventional direct organic-inorganic assembly. Typically, nitrogen-doped mesoporous $\varepsilon$-$WO_3$ with high specific surface area, uniform mesopores and stable framework is obtained and exhibits great application potentials such as gas sensing.

Ordered structures based on molecular assembly widely exist in nature, such as biological membranes, biophotonic structures, diatomite and insect compound eyes[1,2]. Learning from Nature, scientists have devoted great efforts to exploit complex and elaborate artificial nanostructures with tailored functions via assembling molecular or nanoscale units into periodic structures through organic-inorganic co-assembly, including zeolites[3], metal-organic frameworks[4], mesoporous materials[5–8], biomimetic materials[9], and superlattice structures[10]. As a family of nanostructured materials, ordered mesoporous materials, especially transition metal oxides with rich components and attractive properties have attracted ever-growing attention due to their high specific surface areas, uniform pores and adjustable

morphologies and thus have a great potential in catalysis, sensing, energy storage, and conversion[11,12]. Until now, hard-template, soft-template and self-template methods were developed to synthesize mesoporous materials[13–16]. Among them, the soft-template method is of special concern due to its flexibility, tunability, and simplicity in constructing mesostructured materials.

In conventional soft-template synthesis of mesoporous materials, small surfactants or amphiphilic block copolymers can co-assemble with precursors through direct organic-organic or organic-inorganic co-assembly, wherein the host organic structure-directing agents directly co-assemble with guest precursors (e.g., silicate or alumino-silicate, synthetic resol, molecular metal salts) through intermolecular

[1]Department of Chemistry, State Key Laboratory of Molecular Engineering of Polymers, Shanghai Key Laboratory of Molecular Catalysis and Innovative Materials, Collaborative Innovation Center of Chemistry for Energy Material (iChEM), Fudan University, Shanghai 200433, China. [2]State Key Lab of Transducer Technology, Shanghai Institute of Microsystem and Information Technology, Chinese Academy of Sciences, Shanghai 200050, China. [3]Shanghai Key Laboratory of Magnetic Resonance, State Key Laboratory of Precision Spectroscopy, School of Physics and Electronic Science, East China Normal University, Shanghai 200241, China. ✉e-mail: yhdeng@fudan.edu.cn

interactions, such as covalent interactions, hydrogen bonding, π-π stacking and Coulombic interactions[17–19]. These methods strongly rely on the direct host-guest interaction and are usually followed by the crosslinking/condensation process of guest precursors, which require strict control over the synthesis parameters, such as solvent polarity, additive type, temperature and ambient humidity[11,20]. Therefore, these assembly processes lack the flexibility in constructing periodic nanoarchitectures with rich compositions in a simple system.

Polyoxometalates (POMs) are a family of nanosized metal-oxo clusters with protons (denoted as acidic POMs) or ammonium and alkali metals (denoted as non-acidic POMs) as counter-cations. Because acidic POMs are compatible in many assembly processes using polar organic solvents, the assembly of acidic POMs and copolymers has recently emerged as an alternative route to synthesize mesoporous metal oxides (mMOs) based on the Coulombic attraction directly between heteropolyacid anions and protonated copolymer[21,22]. However, this direct Coulombic interaction is unfavorable for the construction of mesostructured materials in a wide synthesis condition. To overcome these limitations and create mMOs with rich compositions and pore structures, it is of great importance and interest to explore a general and versatile copolymers templating route suitable for various POM precursors.

In this study, we use varieties of molecular pairs (e.g., dimethylformamide (DMF)/$H_2O$) as binary solvents, which are completely miscible at the molecular level in a relatively wide mixing ratio range (water content < 20 vol%) and form molecule-like complexes. A general "solvent-pair surfactants" enabled assembly (SPEA) between amphiphilic block copolymers (AB copolymers) and POMs is developed to form ordered mesostructures through non-direct host-guest interaction. Owing to hydrophilic and DMF-phobic characters, Non-acidic POMs can be readily dissolved in the binary solvent by virtue of the solvation effect of DMF·$nH_2O$ complexes, which are analogous to surfactants and thus were called "solvent-pair surfactants". During the solvent evaporation process, POMs are accumulated preferentially at the regions between hydrophilic PEO segments of spherical PEO-$b$-PS micelles mediated by "solvent-pair surfactants", forming ordered POMs/AB copolymers mesostructures. The ordered mesostructures can be maintained during the conversion from POMs to highly crystalline metal oxides via thermal treatment. The counter-cations also participate in the assembly and annealing process, forming heteroatom-doped or alkali metal-intercalated mMOs. As a proof of concept, the SPEA approach can generate ordered mesoporous nitrogen-doped $\varepsilon$-$WO_3$ (denoted as mN-$WO_3$) with uniform spherical pores (12.1–33.2 nm), iso-oriented nanocrystalline walls and homogeneous N-doped framework by using $(NH_4)_6H_2W_{12}O_{40}$ as precursor. Thanks to the highly porous structure and rich active sites, the obtained mN-$WO_3$ semiconductors show superior sensing performances towards trace acetone. Moreover, through combining different precursors (POMs or molecular metal salts) to assemble with different AB copolymers, the SPEA method can be extended readily to design mMOs with adjustable pore sizes and compositions for various applications.

## Results and discussion

### Preparation of assembly system

In order to achieve a flexible co-assembly of AB copolymers with various POM precursors, it is crucial to precisely design an aqueous binary solvent, because most of the AB copolymers and POMs (especially for non-acidic POMs) can be dissolved merely in polar organic solvent and water, respectively[23,24]. Some organic solvents, such as small alcohols and tetrahydrofuran (THF), are apparently miscible with water at any mixing ratio. However, microscopic heterogeneity at the molecular level was found in these mixtures and most of the water molecules exist as small clusters in the organic-rich circumstance due to the remnants of the three-dimensional hydrogen-bonded network

structure of bulk water[25–27]. We found that the introduction of a small amount of non-acidic POMs aqueous solution to these organics causes the naked oleophobic POMs to directly interact with organic solvents due to the molecular aggregation of water, leading to their macroscopic aggregation and precipitation (Supplementary Fig. 1a and b). Owing to the strongly polarizable amido of DMF, the hydrogen bonding between DMF and water is strong enough to completely destroy the hydrogen-bonded network structure of bulk water in the case of low water content (<20 vol%), forming molecule-like DMF·$nH_2O$ complexes[28–31]. These complexes can behave as "solvent-pair surfactants" to solubilize POMs in binary solution and serve as a key medium enabling the hydrophilic interaction between POMs and PEO segments (Supplementary Fig. 1c and d).

Based on experimental results, we classify common organic solvents that can dissolve amphiphilic copolymers into three types (Supplementary Table 1). The first type has molecular structure similar with DMF. Other two types are immiscible with water at the molecular and macroscopic scales, respectively. As expected, by replacing DMF with other solvents, all the first type solvents can be used to design organic solvent/water pairs for co-assembly of non-acidic POMs and PEO-$b$-PS into periodic structures with spherical, cylindrical or lamellar mesopores (Supplementary Fig. 2).

We further investigate the physiochemical difference of these solvents based on density functional theory (DFT) calculations and molecular dynamic (MD) simulations. Quantitative molecular electrostatic potentials and dipole moment values reveal that the first type solvents have larger dipole moments (Supplementary Table 2) and lower the most negative potentials (Supplementary Fig. 3) than water. It indicates these organic solvent molecules and water can form intermolecular interaction which is much stronger than water-water intermolecular interactions. In contrast, the second type solvents, including THF, acetone, and 1,4-dioxane, exhibit the opposite tendency. Taking DMF and THF as models for further exploration, the total interaction energy of DMF-water is evidently lower than THF-water, indicative of a more stable structure of DMF-water complexes (Supplementary Fig. 4 and Table 3). MD simulations show the complete molecular level mixing of DMF and water and no phase separation was observed even at extended simulation timescales, while the water clusters were clearly noticed in THF-water mixing system (Supplementary Fig. 5 and Table 4).

### Formation mechanism

The molecular level mixing of binary solvent and the formation of organic solvent-water complexes with stable structure are essential for the dissolving of non-acidic POMs. POM clusters with oxygen-rich surface and relatively low surface charge density can easily form hydrogen bonding with water, but this interaction is not enough to destroy the DMF-water interactions. Therefore, the homogeneous state of binary solvent is maintained and the DMF-water complexes can spread around the POMs like surfactants, in which the water head interacts with POM and the DMF tail spread out in the continuous organic surroundings[32]. These "solvent-pair surfactants" guarantee the dissolving of POMs prior to the assembly process and serve as a medium to guide the gathering of POMs selectively around the hydrophilic segments of AB copolymers, because of the preferential accumulation of water around hydrophilic blocks (Fig. 1a, b)[33,34]. Based on these useful "solvent-pair surfactants", the counter-cations of metal-oxo clusters can be involved in the assembly process and finally incorporated in the obtained materials after calcination, leading to functional multicomponent mMOs[35]. In comparison, ordinary metal acid ions, such as $WO_4^{2-}$ and $SnO_3^{2-}$, are almost insoluble in DMF-water system, because their high surface charge density can induce too strong interactions with water and break DMF-water complexes, leading to the microscopic phase separation of binary solvents and the precipitation of hydrophilic solute[36–38]. This finding and understanding

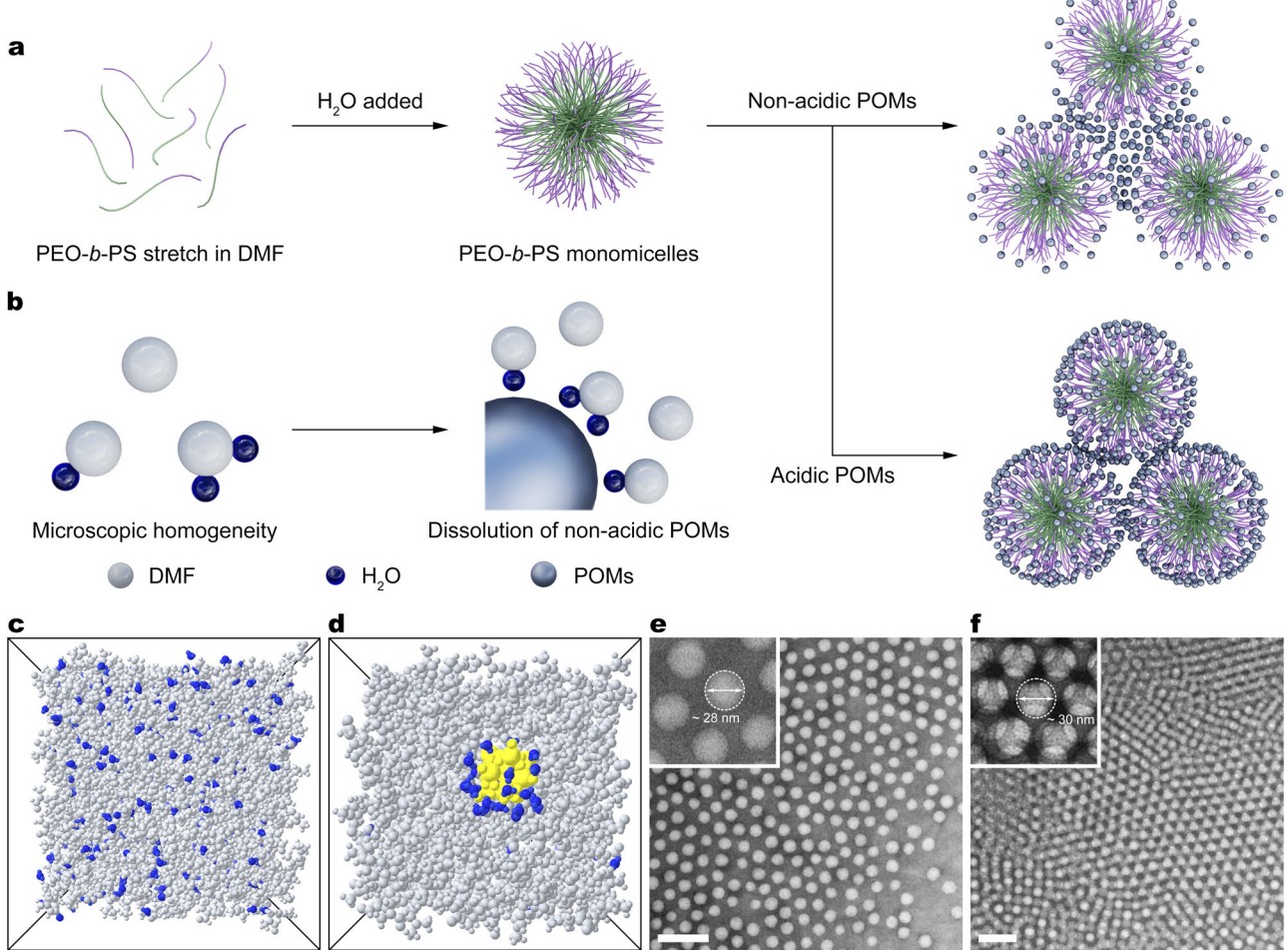

**Fig. 1 | Co-assembly of PEO-*b*-PS and POMs. a, b** Schematic illustration of the PEO-*b*-PS micelles interacting with acidic or non-acidic POMs (**a**) and the micro-environment of non-acidic POMs in assembly system (**b**). **c, d** Snapshots of MD trajectories at 50 ns for DMF-water system (**c**) and $Na_3PW_{12}O_{40}$-DMF-water system (**d**). The gray, blue and yellow regions represent DMF, $H_2O$ and POM, respectively. **e, f** TEM images with higher magnification images of the co-assembly between PEO-*b*-PS micelles and $Na_6H_2W_{12}O_{40}$ with low concentration (**e**) and high concentration (**f**). The scale bars are 100 nm.

about the subtle and robust balance between solute and solvent-pair provide us sparkle to design and regulate the multicomponent assembly system.

Thus, in this study we proposed and verified a "solvent-pair surfactants" enabled assembly (SPEA) toward the controlled organization of non-acidic POMs and AB copolymers (e.g., poly(ethylene)-*block*-polystyrene, abbreviated as PEO-*b*-PS) in DMF/$H_2O$ binary solvent. In this system, DMF-insoluble non-acidic POMs can be readily dissolved in DMF solvent by means of the solvation effect of DMF·$nH_2O$ complexes, which is confirmed by MD simulations (Fig. 1c, d). Upon solvent evaporation, PEO-*b*-PS micelles stack up into close-packed liquid-crystal structures on the substrates. Meanwhile, with the increase of concentration, hydrophilic POMs were confined and aggregate around the hydrophilic PEO segments of PEO-*b*-PS with the mediation of DMF·$nH_2O$ solvent pair surfactants. After complete drying, stable POMs/PEO-*b*-PS ordered mesostructures can be obtained. Fourier transform infrared (FTIR) and UV-vis absorption spectra confirm that POMs have no direct strong intermolecular interactions with PEO-*b*-PS (Supplementary Figs. 6 and 7). [1]H nuclear magnetic resonance spectra suggest that PEO and POMs simultaneously interact with $H_2O$ through hydrogen bonding in the DMF/$H_2O$ binary solvent system with appropriate compositions, implying that POMs interact with PEO-*b*-PS through the mediation of $H_2O$ or DMF·$nH_2O$ complexes (Supplementary Figs. 8–10)[34,39]. In principle, the "solvent-pair surfactants" concept is applicable for all POMs which are easily soluble in water and insoluble in DMF. Following this principle, the SPEA method was successfully used to realize the co-assembly of PEO-*b*-PS with nine kinds of POMs composed of different transition metals and counter-cations (Supplementary Figs. 11 and 12), such as $(NH_4)_6H_2W_{12}O_{40}$, $Na_4V_{10}O_{28}$, and $K_3Mo_6O_{19}$, implying a general assembly process.

Through mixing PEO-*b*-PS/DMF solution with non-acidic POMs aqueous solution, the PEO-*b*-PS underwent a microphase separation to form spherical micelles containing PS cores and PEO shells. Due to the existence of water, which is a poor solvent for PS, the movement of hydrophobic PS block was frozen, forming stable colloidal solution of spherical micelles and dissociative POMs[40]. Transmission electron microscopy (TEM) characterization shows that after directly dropping the colloidal solution on the carbon-coated copper grids, numerous bright spherical micelles can be observed and the space around the PEO region is decorated (Fig. 1e) or filled with dark POMs (Fig. 1f), confirming the selective aggregation tendency of POMs.

By employing a two-step sequential thermal treatment in nitrogen and air[41], three kinds of nitrogen-doped mMOs and six kinds of multimetallic mMOs with high crystallinity and ordered mesopores were obtained (Fig. 2a). Thermogravimetric analysis (TGA) and X-ray diffraction (XRD) images confirm that the POMs were converted into metal oxides at high temperatures (Supplementary Figs. 13–15). Field emission scanning electron microscopy (FESEM) observation reveals

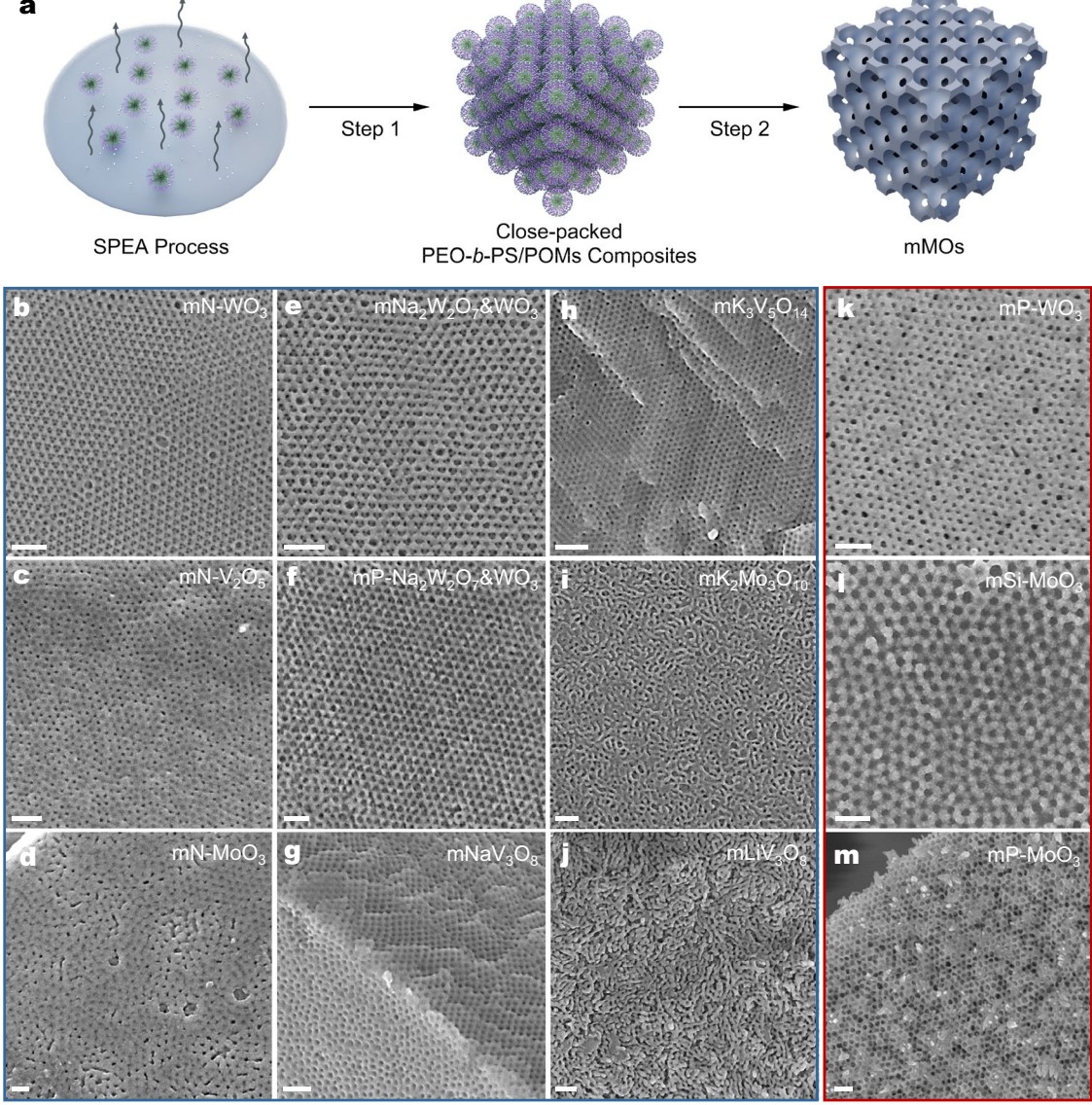

**Fig. 2 | Library of mMOs constructed by PEO-*b*-PS and POMs. a** Schematic illustration of the synthesis process of mMOs. Step 1: the spherical PEO-*b*-PS micelles and POMs packed into ordered *fcc* mesostructure through the "solvent-pair surfactants" enabled assembly. Step 2: mMOs were obtained by calcining in N₂ with the pyrolysis of PEO-*b*-PS and decomposition of POMs, and then in air to remove the carbon. **b**–**m** mMOs are prepared with non-acidic POMs (**b**–**j**) and acidic POMs (**k**–**m**). The scale bars are 100 nm.

that the products have ordered mesoporous structures with uniform pores (Fig. 2b–h). The mesoporous $K_2Mo_3O_{10}$ and $LiV_3O_8$ materials possess short-range order due to the limited solubility of their precursors and the fast crystallization rate (Fig. 2i, j, Supplementary Fig. 16). X-ray photoelectron spectroscopy (XPS) and Energy-dispersive X-ray (EDX) element mapping analysis show a homogeneous element distribution of products, indicative of a full conversion of the precursors into corresponding oxides (Supplementary Figs. 17–24).

In addition to non-acidic POMs, acidic POMs were also used in this aqueous binary solvent assembly system, while the driving force is dominated by the Coulombic attraction rather than SPEA process. The PEO segments of PEO-*b*-PS AB copolymers can be quickly protonated by H⁺ ions released from POMs and further interact with POM anions by Coulombic attraction, forming PEO-*b*-PS/POMs composite micelles[22] (Fig. 1a, Supplementary Fig. 9j–l). The subsequent solvent evaporation can induce the composite micelles to pack closely into ordered mesostructures. Similarly, using acidic POMs ($H_3PW_{12}O_{40}$, $H_4SiMo_{12}O_{40}$, $H_3PMo_{12}O_{40}$) as precursors, three kinds of heteroatom-

doped mMOs can be readily obtained after thermal treatment, respectively (Fig. 2k–m, Supplementary Figs. 25–27).

It is noteworthy that, because the SPEA method does not rely on the direct interaction between PEO-*b*-PS and POMs, it can be applicable for other amphiphilic copolymers, including electroneutral poly(ethylene oxide)-*b*-poly(methyl methacrylate) (PEO-*b*-PMMA), poly(ethyl oxide)-*b*-polybutylene (PEO-*b*-PB), positively charged poly (4-vinylpyridine)-*b*-polystyrene (P4VP-*b*-PS) and negatively charged poly(acrylic acid)-*b*-polystyrene (PAA-*b*-PS). All these AB copolymers were demonstrated to be effective in synthesizing mMOs via the aqueous binary solvent assembly system (Supplementary Fig. 28). These results confirm that this "solvent-pair surfactants" concept enables a flexible synthesis of mMOs (Fig. 3).

**Structure and characterization of mN-WO₃**
Taking the synthesis of mN-WO₃ using $(NH_4)_6H_2W_{12}O_{40}$ and PEO₁₁₄-*b*-PS₂₀₀ in DMF/H₂O system as an example, the as-made PEO-*b*-PS/ $(NH_4)_6H_2W_{12}O_{40}$ mesostructured composite was calcined at 500 °C in nitrogen and then at 400 °C in air to form crystalline metal oxide

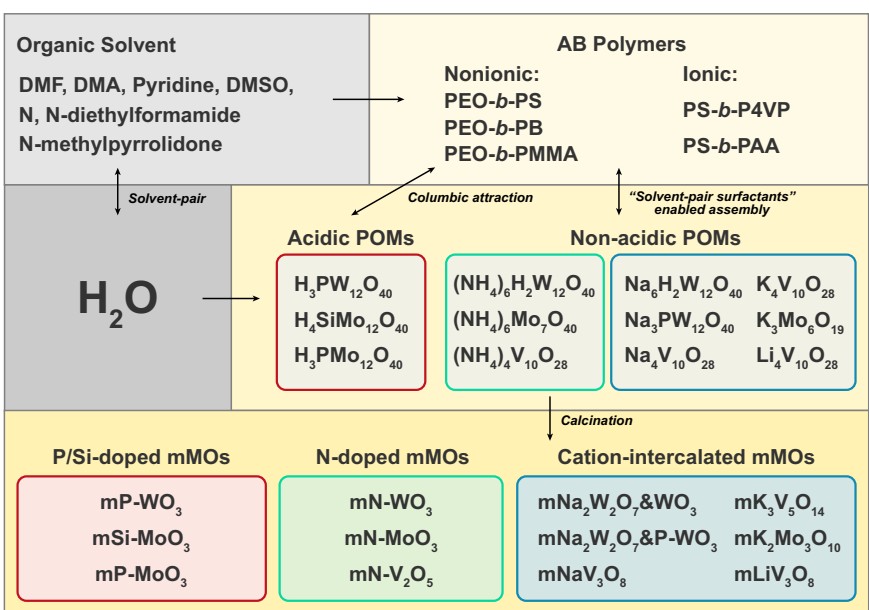

**Fig. 3 | Summary of the generalized synthesis of mMOs in aqueous binary solvent system based on POMs precursors.** All the contents are covered in this study.

framework and remove PEO-*b*-PS. The obtained mN-WO₃ retained the ordered mesostructure (Supplementary Figs. 29–31). FESEM and TEM observations reveal that the obtained mN-WO₃ exhibit an *fcc* arranged mesoporous structure with inter-connected pores regularly aligned over large domains (Fig. 4a, b), which was further confirmed by small-angle X-ray scattering (Supplementary Fig. 32). Nitrogen adsorption-desorption isotherms of the mN-WO₃ show typical type-IV curves with an H₂ᵦ-type hysteresis loop, indicative of spherical mesopores with large windows. The pore size distribution shows a uniform pore size distribution around 20.9 nm. By increasing the repeating units of PS segments of PEO-*b*-PS from 103 to 275, the pore sizes can be tuned in the range of 12.1–33.2 nm and pore structures can be adjusted from spherical, cylindrical to lamellar (Supplementary Figs. 33–35).

High resolution transmission electron microscopy (HRTEM) image reveals that the mN-WO₃ has iso-oriented nanocrystalline walls (Fig. 4c, d). Selected-area electron diffraction (SAED) pattern shows a single-crystal-like diffraction spot[42,43] (inset in Fig. 4b). The iso-oriented crystalline walls endow the mesostructure of mN-WO₃ ultrahigh thermal stability even at 700 °C. For comparison, mesoporous WO₃ (i.e., mWO₃) was synthesized by using WCl₆ as precursor following a conventional evaporation-induced co-assembly process[44]. It has polycrystalline walls which underwent a partial collapse at 700 °C in nitrogen (Supplementary Figs. 36–40). It is speculated that the formation of iso-oriented walls originates from the crystalline-crystalline transition from POMs to WO₃ through the reconstruction of W-O octahedrons, while amorphous-crystalline transition occurs in the case of mWO₃[45,46].

Moreover, the ammonium counter-cations of (NH₄)₆H₂W₁₂O₄₀ precursor can be involved in the SPEA process, yielding nitrogen-doped mWO₃ after the thermal treatment. EDX element mapping analysis shows the uniform distribution of W, O, and N throughout the mesoporous framework, indicating a homogenous doping of nitrogen (Fig. 4e). No characteristic peaks of N-H were found in FTIR spectrum of the mN-WO₃, thus the N 1*s* peak of XPS at 400 eV is ascribed to W-O-N (Fig. 4f, Supplementary Figs. 41 and 42a)[47]. The well-resolved XRD pattern of mN-WO₃ is ascribed to the unusual *ε*-WO₃ crystallites, while the mWO₃ synthesized using WCl₆ precursor shows a *γ*-phase, and the results agree well with Raman spectroscopy measurements (Fig. 4g, Supplementary Fig. 42b). The formation of ferroelectric *ε*-WO₃ is considered as the lattice distortion of *γ*-phase caused by the interstitial doping of nitrogen atoms in WO₃ lattice[22]. Besides, UV-vis diffuse

reflectance spectra reveal the doping of nitrogen causes the reduction of band gap from 2.70 to 2.63 eV because its *p* states mix with O 2*p* states[48] (Supplementary Fig. 43). Synchrotron-radiation-based X-ray absorption fine structure spectroscopy is applied to resolve the crystal structure of mN-WO₃ directly. The W L₃-edge extended X-ray absorption fine structure reveals that the W-O coordination number of mN-WO₃ and mWO₃ have no significant difference (Supplementary Fig. 44 and Table 5), while the local atomic and electronic structure of them differ obviously (Fig. 4h), which can be attributed to the introduction of nitrogen atoms. Density functional theory (DFT) calculations indicate that N-interstitial doping is an optimal doping type (Supplementary Figs. 45 and 46), which is consistent of the aforementioned results by XPS and XRD analysis.

Owing to the highly ordered and inter-connective mesopores, iso-oriented crystalline walls, and highly polarized surface, it is expected to find use in wide range of applications for mN-WO₃. For example, because it is favorable for gas diffusion[49], electron transport, and selective adsorption of polar gases, mN-WO₃ shows superior gas-sensing performances towards acetone with high sensitivity ($S = 103.7$ for 50 ppm) even at ppb-level ($S = 1.9$ for 10 ppb), good selectivity and fast response-recovery dynamics (15/35 s) (Supplementary Figs. 47–52 and Table 6). For comparison, the gas-sensing performances of aforementioned mWO₃ were investigated, while its response rate is much slower and response value is much lower than mN-WO₃, due to the relatively weak interactions between acetone and mWO₃ (Fig. 4i).

## The generality of the assembly approach

As described above, almost all the water-soluble POMs can be fabricated into ordered mesostructures and then converted into metal oxides. The compositions of the obtained metal oxide mesostructures depend on both metal-oxo clusters and their counter-cations (not limited to the species we investigated). The pore structures can be adjusted from spherical (Fig. 5a), cylindrical (Fig. 5b) to lamellar (Fig. 5c) by changing the hydrophobic segment length of the AB copolymers and tuning the dynamic equilibrium of the composite micelles under specific conditions. Moreover, because this SPEA approach does not rely on direct host-guest interactions, various POMs with different compositions can be simultaneously used in this assembly system without interference. The obtained corresponding mMOs have ordered structures and uniformly distributed elements (Supplementary Figs. 53–55).

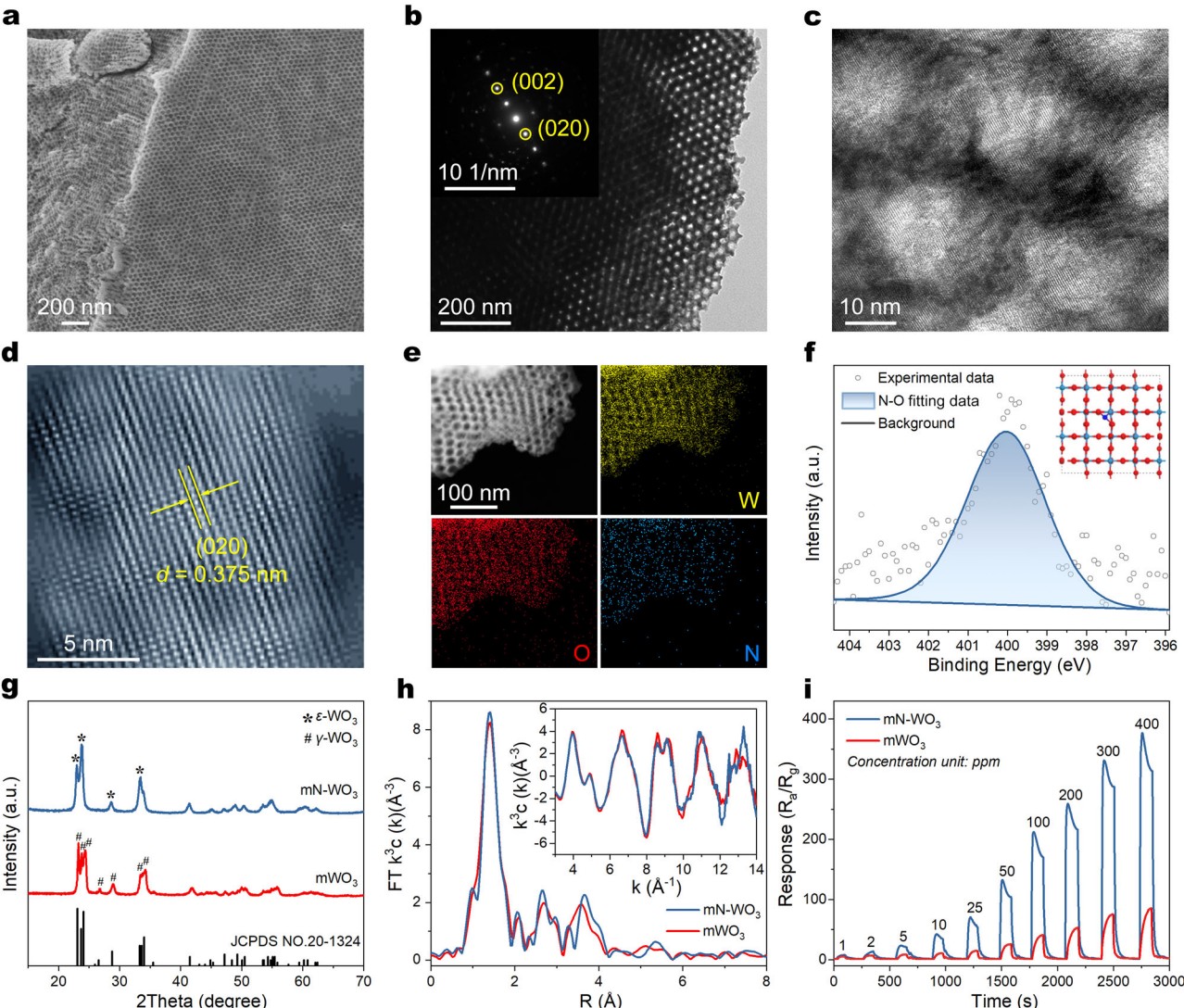

**Fig. 4 | Structural and elemental characterization of ordered mN-WO₃.**
**a**, **b** Typical FESEM (**a**) and TEM images with a selected-area electron diffraction pattern (**b**) of ordered mN-WO₃. **c** HRTEM image of mN-WO₃, indicative of iso-oriented crystalline walls. **d** ac-HAADF-STEM image of mN-WO₃ produced with fast Fourier transform (FFT) and then reverse FFT. **e** EDX element mapping analysis of the structure. **f** XPS spectrum shows the N 1$s$ core level peak region. The inset is structural model of N-WO₃ constructed by DFT calculations, revealing the interstitial doping of nitrogen. **g** XRD patterns of mN-WO₃ and mWO₃ (the control sample). Both of them are monoclinic WO₃, while the refined structure of mN-WO₃ is attributed to the $\varepsilon$-phase and the mWO₃ is attributed to the $\gamma$-phase. **h** Fourier transform W $L_3$-edge EXAFS spectra of the samples and the inset is W $L_3$-edge EXAFS oscillation function $k^3c(k)$. **i** Response-recovery curves of gas sensors based on mN-WO₃ (blue line) and mWO₃ (red line) under different acetone concentrations at 300 °C.

Due to the unique "solvent-pair surfactants" enabled indirect interaction, the SPEA method allows for a facile strategy to synthesize functionalized mMOs with tailored pore-wall microenvironments by directly introducing additional components like precursors of other metal oxides and noble metal nanoparticles. A series of heteroatom-doped mMOs (Fig. 5d, Supplementary Figs. 56–58), metal oxide composites (Fig. 5e, Supplementary Fig. 59) and noble metal-loaded mMOs (Fig. 5f, Supplementary Fig. 60) can be obtained, which are highly desired in various fields such as catalysis and energy. The generality of this SPEA approach in constructing mMOs with diverse structures and compositions is summarized in Fig. 5.

In summary, a "solvent-pair surfactants" concept was proposed to realize an unconventional organic-inorganic molecular assembly in constructing functional mesostructured materials, especially the indirect co-assembly of POMs and AB copolymers. The organics-water pairs in aqueous binary solvent solution were found to act as surfactants to allocate the inorganic species at the water-swelled PEO, PAA or P4VP region of AB copolymer micelles, which enables the formation of ordered mesostructures as the solvent evaporates and the derived mMOs after thermal treatment. This SPEA approach was demonstrated to be capable of synthesizing a library of mMOs, which is highly desired in catalysis and chemical sensing, in various organics-based solvent-water pair systems using different POMs and AB copolymers. This study provides a route of molecular assembly towards functional nanostructures and their derived functional porous materials with rich compositions and structures. It is conceivable that sub-nanometer units with features like POMs, such as metal-sulfur clusters and quantum dots, can also be used to construct ordered mesostructures on the basis of this SPEA approach. It also sheds light on the precise design and understanding of supramolecular self-assembly, chemical synthesis, and complex biological systems.

## Methods

### Chemicals and materials

Ammonium metatungstate hydrate ((NH₄)₆H₂W₁₂O₄₀·xH₂O, 99.5% metals basis), methoxypolyethylene glycol (PEO, M_w = 5000 g/mol)

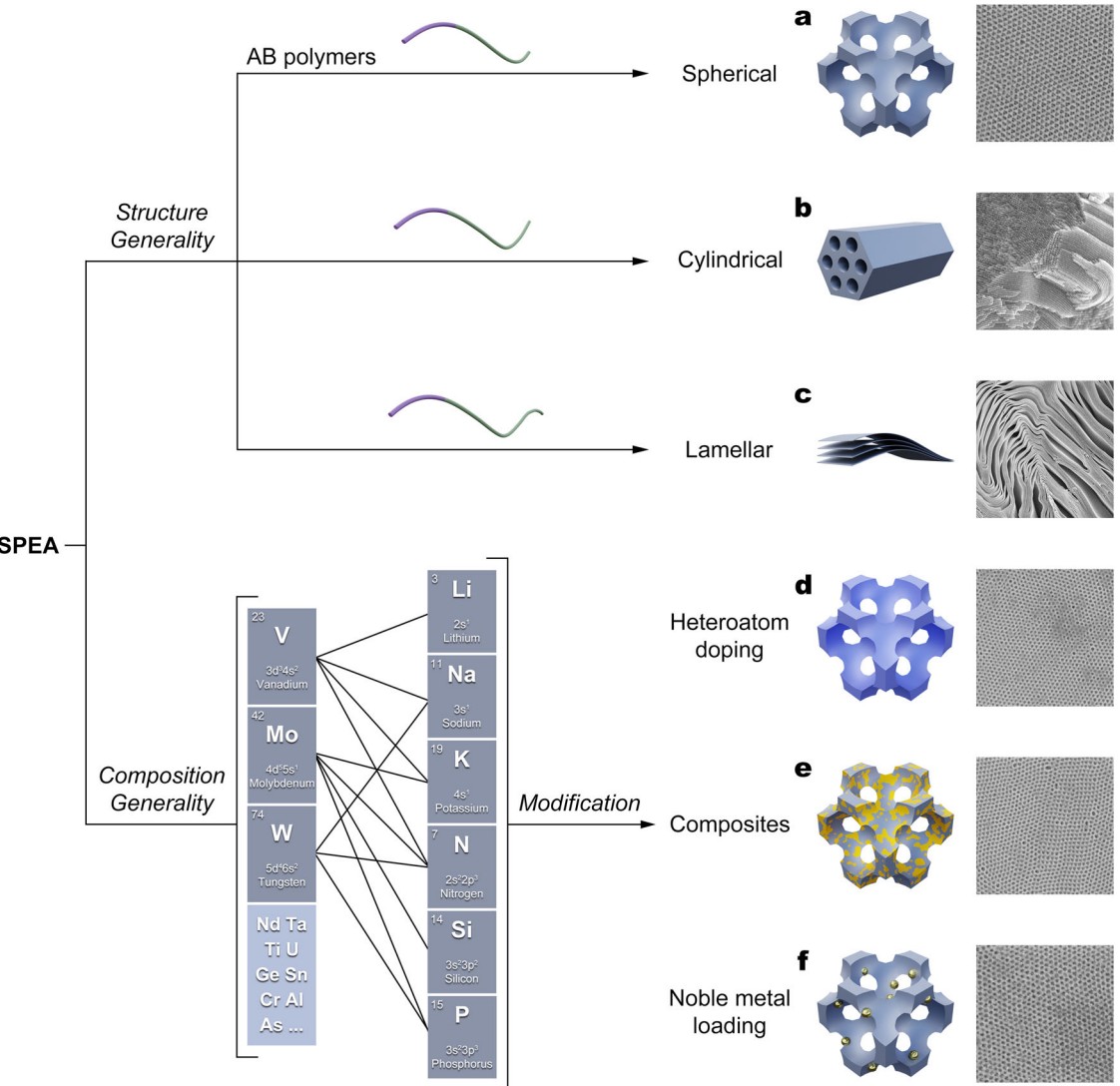

**Fig. 5 | Schematic illustration of generality of the SPEA approach in constructing mMOs with diverse structures and compositions.** The matched element groups at the left bottom show the combination of elements covered in this study and unmatched groups show other possible combinations. **a**–**f** Schematic structures and corresponding SEM images of mMOs with spherical (**a**), cylindrical (**b**), lamellar (**c**) pore structures and heteroatom-doped (**d**), composite (**e**), noble metal loaded (**f**) compositions, respectively.

and deuterated N, N-dimethylformamide (DMF-d$_7$, 99.5%) were purchased from Aladdin. N, N-dimethylamide (DMF, ≥ 99.5%) and styrene (C$_8$H$_8$, AR) was purchased from Sino-Pharm Chemical Reagent Co. Ltd.

### Synthesis of ordered mesoporous nitrogen-doped WO$_3$ (mN-WO$_3$)

0.05 g of amphiphilic diblock copolymer PEO$_{114}$-b-PS$_{200}$ (M$_w$ = 25800 g/mol, polydispersity index = 1.08) was dissolved in 4.5 mL of DMF to form a homogeneous solution. Meanwhile, 0.15 g of (NH$_4$)$_6$H$_2$W$_{12}$O$_{40}$ was dissolved in 0.5 mL of water. Gradually adding the (NH$_4$)$_6$H$_2$W$_{12}$O$_{40}$ water solution to copolymer DMF solution with a high-speed mixer, a light-blue colloidal solution (the mass ratio of PEO-b-PS/POMs was 1:3) was obtained. After stirring for 1 h, the colloidal solution was cast on glass Petri dished to evaporate DMF and water at 40 °C for 24 h, followed by further heating in an oven at 100 °C for 24 h to solidify the structure. Then, the obtained PEO-b-PS/POMs film was calcined at 500 °C for 0.5 h in N$_2$ (heating rate, 1 °C/min below 350 °C and 5 °C/min above 350 °C) and then at 400 °C for 0.5 h in air (5 °C/min) to produce ordered mN-WO$_3$.

### Characterizations and measurements

Field-emission scanning electron microscopy (FESEM) images were collected with Zeiss GM500 FESEM (Germany) performed at 3 kV. High contrast transmission electron microscopy (HCTEM) images were recorded with Hitachi HT7800 (Japan) at 100 kV. High-resolution TEM images were obtained at a Tecnai G2 F20 S-Twin field emission transmission electron microscopy (FEI, America) at 200 kV. Aberration-corrected high angle annular dark-field scanning transmission electron microscopy (ac-HAADF-STEM) images were recorded on FEI-Titan Cubed Themis G2 300 (the Netherlands). Powder X-Ray diffraction (XRD) patterns were obtained at a Bruker D4 X-ray diffractometer (Germany) equipped with Ni-filtered Cu Kα radiation (40 kV, 40 mA). X-ray photoelectron spectroscopy (XPS) data was collected with an AXIS ULTRA DLD XPS System with MONO Al source (Shimadzu Corp., Japan). N2 adsorption-desorption isotherms were obtained at 77 K with an ASAP 2420 analyzer. Before measurements, the samples were degassed in vacuum at 180 °C for 6 h to remove impurities. The specific surface area was recorded with Brunauer-Emmett-Teller (BET) method and the size distributions were dealt with Barrett-Joyner-Halenda (BJH) model. Thermogravimetric analysis data was obtained

from 25 to 700 °C in air with a heating rate of 5 °C/min. Fourier transform infrared spectra (FTIR) of samples were collected with a Nicolet Fourier spectrophotometer. Proton nuclear magnetic resonance (NMR) spectrum was recorded with 500 MHz Bruker AVANCE III HD spectrometer (Germany).

## Gas sensing tests

The gas sensing performances of all the samples were fabricated into side-heated type of gas sensor and evaluated on MA1.0 gas sensing measuring system (Narui Corp. Ltd., China). The mMOs powders were ground and dispersed into ethanol, and the obtained dispersion was uniformly deposited on alumina tubes. The distance between the two Au electrodes on alumina tubes was 2.0 mm on average. The as-fabricated tubes with deposited samples were then subjected to annealing at 100 °C for 2 h and then at 300 °C for 48 h to ensure the stability of the devices. The target liquid analytes were injected onto an evaporation plate to generate target gases. The response time and recovery time are the times required to achieve a 90% saturation value after the import and release of the target gas within the step function.

## Reporting summary

Further information on research design is available in the Nature Portfolio Reporting Summary linked to this article.

## Data availability

The data generated in this study are provided in the Source Data file. Source data are provided with this paper.

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

## Acknowledgements

This work was financially supported by National Natural Science Foundation of China (22125501 Y.D., U22A20152 Y.D., 22105043 Y.Z.), Key Basic Research Program of Science and Technology Commission of Shanghai Municipality (20JC1415300 Y.D.), State Key Laboratory of Transducer Technology of China (SKT2207 Y.D.), China Postdoctoral Science Foundation (2021TQ0066 Y.Z., 2021M690660 Y.Z.) and Fundamental Research Funds for the Central Universities (20720220010 Y.D.).

## Author contributions

W.X., Y.R., and Y.D. conceived the project and designed the experiments. W.X., F.J., B.Y., and Y.D. were primarily responsible for the data collection and analysis. W.X., X.H., Y.Z., and Y.D. prepared the figures and wrote the main manuscript text. W.X., Y.R., J. Li and K.C. analyzed the mechanism with DFT calculations. J. Liu and B.H. were responsible for the NMR characterization and analysis. All the authors contributed to the discussions and manuscript preparation.

## Competing interests

The authors declare no competing interests.
