## [Peer Review File · Nature Communications]

Solvent-Pair Surfactants Enabled Assembly of Clusters and Copolymers towards Programmed Mesoporous Metal OxidesReviewers' Comments:

Reviewer #1:

Remarks to the Author:

In this manuscript, the authors present a novel synthesis method called "solvent-pair surfactants" enabled assembly (SPEA) for the production of mesostructured materials requiring no direct host-guest interactions. The study highlights the merits of the SPEA method by demonstrating that the resulting material exhibits iso-oriented crystalline walls and enables successful counter-cation impregnation. Furthermore, the authors showcase the versatility of this method by generating a wide range of mesostructured materials with diverse compositions, using polyoxometalates (POMs). Overall, the material characterization and formation mechanism have been comprehensively investigated, and the efficacy of the fabricated material has been well demonstrated. Therefore, I recommend the publication of this report. However, there are several minor issues that need to be addressed prior to publication.

1. Please check that the supporting information is appropriately numbered in the order referenced in the text and that it is adequately mentioned.
2. In Fig. 1e, the authors mention that the space surrounding the poly(ethylene oxide) (PEO) region is decorated. It is suggested to include an inset of a higher magnification image to provide clearer visualization.
3. In Fig. S31-33, the pore size distribution curve indicates an increase in pore size as the polystyrene (PS) chain lengthens. However, this trend contradicts the observations in the SEM image. Please review the image to address this discrepancy.
4. The authors assert that mN-WO₃ exhibits thermal stability even at 800 °C and provide XRD data as evidence. To strengthen this claim, it is recommended to include additional TEM or SEM images demonstrating the preservation of its structural integrity.
5. The authors mention the possibility of obtaining heteroatom-doped mMOs, metal oxide composites, and noble metal-loaded mMOs. Please include the synthesis methods for these materials in the Supplementary Information section.

Reviewer #2:

Remarks to the Author:

In this work, authors prepared a range of mesoporous metal oxides with a method termed SPEA utilizing solvent-pair surfactant concept. This work is extensively conducted with DFT calculations, molecular simulations, experimental analysis and characterizations, and discussion is well-written. It can be considered towards acceptance after addressing the following comments.

1. Figures must be discussed in order. (e.g., Fig. 1c is discussed before Fig. 1a and b, and many other misordering of the figure number and alphabets in the manuscript).
2. The demonstrated method (SPEA) seems quite similar to the existing methods utilizing soft-templates (e.g., EISA). Further discussion on different types of existing similar synthetic methods should be made, and the significance of the current SPEA method should be extensively highlighted.
3. Although DFT calculation-based discussion is present, experimental discussions on the effect of binary systems with other organic compounds should be added.
4. 'Columbic attraction' should be revised to 'Coulombic attraction'
5. Figure resolution must be improved, especially with the schematics.
6. Fig. 2 i and j seem to demonstrate cylindrical mesochannels rather than mesopores. Their TEM

images should be supplied to confirm the nature of their porosity.

7. Fig. 3 should be revised to what is relevant to and covered in this study (as far as I understand, this study investigated DMF and PEO-b-PS only among the presented organic solvents and ABCs).

8. As this work mainly focuses on the novel synthesis of porous materials, it would benefit a broader readership to introduce porous materials prepared with other types of methods (e.g., Nature protocols 17 (12), 2990-3027; Chemical Science 13 (36), 10836-10845; Angewandte Chemie International Edition 60 (51), 26528-26534).

Reviewer #3:

Remarks to the Author:

The work by Xie et al. reports a facile and flexible "solvent-pair surfactants" enabled assembly strategy to synthesize a series of mesoporous metal oxides based on co-assembly of POM clusters and block copolymers. The developed method exhibits unique advantages over conventional EISA process, including the interesting process involving the smart and indirect assembly of polyoxometalates with block copolymers, as it exhibits greater flexibility in terms of compositions and synthesis conditions. The authors also offer new conceptual understanding for the underlying principles about dissolution and assembly process of metal-oxo clusters in binary solvents. These results are interesting and will be great help to materials scientists and chemists working in related fields. The as-designed materials show great application potential in catalysis and chemical sensing. From this point of view, I believe this paper can be published on Nature Communications after minor revisions.

1. In page 2, line 1, "uniform mesopores and stable framework was obtained and exhibits great application potentials such as gas sensing", the usage of tenses should be unified.
2. Figure 2 inside notes, "calcination" is incorrectly used.
3. The scale bars in Figure 2d, g, j and m are missing.
4. In Figure S10d-g, the baseline keeps rising as the increase of temperature after the thermal degradation stage. There is also an abnormal fluctuation in Figure S10k, 250-300 °C stage. The authors should explain these data, or repeat the TG tests.
5. TG data of the PEO-b-PS/AMT should be given to see the decomposition behaviors of copolymer templates.
6. How about the morphology of the mesoporous nitrogen-doped WO₃ after the gas sensing tests, please check if the ordered structure is retained well during the high temperature test processes.
7. Why the gas sensing performances of mesoporous WO₃ prepared by conventional EISA method is inferior to mesoporous nitrogen-doped WO₃? The authors should give more discussions about the comparison of these two kinds of mesoporous materials.
8. In Table S7, the comparative literature needs to be updated.
9. For the multicomponent co-assembly process, how did the author guarantee that the noble metal nanoparticles loaded on the surface of mesopores, instead of embedding inside the WO₃ matrix. If it is the latter case, the integration of noble metal and WO₃ doesn't seem to make much sense. The authors should give more discussions on it.
10. In Figure 5, left bottom, some elements are unmatched. The authors should explain the meaning of these matched and unmatched element groups.

Reviewer #4:

Remarks to the Author:

The article reports the well ordered synthesis of a several metal oxides based on POMs assembled with diblock polymers (BCP) and DMF/water mixtures. While POM/BCP assemblies are not novel alone (DOI: 10.1039/C3TA10333A), the main claim of this manuscript is the "solvent pair surfactants enabled assembly" concept which was suggested to operate without direct interaction of the POM with

the BCP. Such non-direct interactions have been widely reported with surfactant based assemblies using counter ion mediation (DOI: 10.1038/368317a0).

The following issues were of significant concern:

1) It seems that the reported data are consistent with traditional polymer structure directing agent approaches (DOI: 10.1557/s43578-021-00421-0) where the DMF/water mixtures are simply a solvent-system optimization to enable solubility of all species. How can the claimed "SPEA" mechanism be distinguished from standard coassembly behaviors? Would not one anticipate hydrophilic polymers to naturally interact with the surfaces of POMs so long as they were well solubilized by the solvent mixture? From this perspective, the main claimed mechanism is not supported.

2) The generalizability was not particularly compelling since all of the BCPs used had generic hydrophilic blocks (PEO, PAA, or P4VP), each of which have previously been demonstrated with BCP SDAs using sol-gel routes. I.e. a suitable solvent system for POMs would have been anticipated to yield such assemblies with any generic amphiphilic SDA. This returns to the point of the "SPEA" mechanism not being directly supported. Here the results are both understandable and expected from standard hydrophilic interactions of POMs with SDAs.

3) The authors claim that iso-oriented walls cause high temperature stability. That claim would need to be supported with evidence such as an otherwise identical sample with non-oriented walls which exhibited less temperature stability. Rather than a real claim, this appears to be a random observation where others have also seem iso-oriented samples previously from SDAs (DOI: 10.1038/NMAT2612).

Additionally, the following aspects were of minor concern:

4) The authors claimed tailored pore size, structure, and pore-wall environment. Were those really tailored by choice, if so what were the guiding principles? Rather it appears that these were a collection of arbitrary observations expected with "dynamic micelle templates".

5) The authors claimed that traditional EISA approaches do not enable tunable structures which is quite incorrect (DOI: 10.1021/cm011209u)

6) The authors seem confused about polymer nomenclature and routinely call diblock polymers "ABCs." That term is reserved most properly for triblock terpolymers whereas all of the polymers used here are more accurately called AB polymers.

7) The authors claim that "different types of direct host-guest interactions may interfere with each other and lead to macroscopic phase separation." The authors should be more clear in what they are trying to say as these words as-written appear self-contradictory.

RESPONSE TO REVIEWERS' COMMENTS

General response: We sincerely thank all the reviewers for their valuable and positive comments, which help a lot to improve the quality of our manuscript. The reviewer comments are in italic font below, and specific concerns are numbered. Our response is given in normal blue font, and all the changes/additions in the revised manuscript are highlighted in yellow.

Reviewer #1

Comments to the Author:

In this manuscript, the authors present a novel synthesis method called "solvent-pair surfactants" enabled assembly (SPEA) for the production of mesostructured materials requiring no direct host-guest interactions. The study highlights the merits of the SPEA method by demonstrating that the resulting material exhibits iso-oriented crystalline walls and enables successful counter-cation impregnation. Furthermore, the authors showcase the versatility of this method by generating a wide range of mesostructured materials with diverse compositions, using polyoxometalates (POMs). Overall, the material characterization and formation mechanism have been comprehensively investigated, and the efficacy of the fabricated material has been well demonstrated. Therefore, I recommend the publication of this report. However, there are several minor issues that need to be addressed prior to publication.

Response: We thank the reviewer for the positive comments.

1. Please check that the supporting information is appropriately numbered in the order referenced in the text and that it is adequately mentioned.

Response: We thank the reviewer's careful reviewing and good suggestion. We reorganized the referenced order of supplementary information. Sections that are not logically related to the main text were deleted (the original supplementary sections 6, 10 and 12) and the content of original supplementary section 9 was moved to the

experimental section. The adjusted referenced supplementary information was highlighted in the revised manuscript.

2. In Fig. 1e, the authors mention that the space surrounding the poly(ethylene oxide) (PEO) region is decorated. It is suggested to include an inset of a higher magnification image to provide clearer visualization.

Response: We thank the reviewer for the good suggestion. We have included the magnified images inset in Fig. 1e and f. Although the POMs cannot be clearly distinguished due to their ultrasmall size (~ 1 nm), the dark region in TEM images could be defined as POMs which is similar to the heteropolyacid staining process before characterization of biopolymers or organic polymers, as proved by previous reports (DOI: 10.1038/s41563-019-0542-x; 10.1002/adma.202100820).

Figure R1. TEM images with higher magnification images of the co-assembly between PEO-*b*-PS micelles and Na₆H₂W₁₂O₄₀ with low concentration (a) and high concentration (b). The scale bars are 100 nm.

3. In Fig. S31-33, the pore size distribution curve indicates an increase in pore size as the polystyrene (PS) chain lengthens. However, this trend contradicts the observations in the SEM image. Please review the image to address this discrepancy.

Response: We thank the reviewer's careful reviewing and good suggestion. We carefully checked and recalibrated the scale bar in Fig. S29c (original Fig. S31) and kept the length of scale bars in Fig.S29a-c consistent.

Figure R2. SEM images of mN-WO₃ synthesized by using PEO-*b*-PS with different molecular weights as templates. (a) PEO₁₁₄-*b*-PS₁₀₃, (b) PEO₁₁₄-*b*-PS₁₇₅ and (c) PEO₁₁₄-*b*-PS₂₅₀.

4. The authors assert that mN-WO₃ exhibits thermal stability even at 800 °C and provide XRD data as evidence. To strengthen this claim, it is recommended to include additional TEM or SEM images demonstrating the preservation of its structural integrity.

Response: We thank the reviewer’s careful reviewing and good suggestion. We are sorry to have a slip here. The “800 °C” should be changed to “700 °C”. We have corrected it in the revised manuscript, see **page 11, line 11**. At 800 °C, the phase change from WO₃ to WO₂ inevitably causes the collapse of pore structures and forms WO₂ rods, see **Fig. S33d**. Below this temperature, mN-WO₃ maintains an ordered structure. TEM and SEM images of mN-WO₃ and mWO₃ after the thermal treatment at 700 °C are shown in **Fig. S34**. The corresponding SAXS data is shown in **Fig. S35**.

Figure R3. SEM images of mN-WO₃ synthesized by using AMT as tungsten precursor after the thermal treatment at 500 °C (a), 600 °C (b), 750 °C (c) and 800 °C (d) in N₂ for 0.5 h.

Figure R4. (a) SEM and (c) TEM images of mN-WO₃ synthesized by using AMT as tungsten precursor after the thermal treatment at 700 °C in N₂ for 0.5 h. (b) SEM and (d) TEM images of mWO₃ synthesized by using WCl₆ as tungsten precursor after the thermal treatment at 700 °C in N₂ for 0.5 h. The scale bars are 100 nm.

Figure R5. Small angle X-ray scattering spectra of the mN-WO₃ and mWO₃ after the thermal treatment at 700 °C in N₂ for 0.5 h.

5. The authors mention the possibility of obtaining heteroatom-doped mMOs, metal oxide composites, and noble metal-loaded mMOs. Please include the synthesis methods for these materials in the Supplementary Information section.

Response: We thank the reviewer's good suggestion. We have included the relevant synthesis methods in the experimental section of supplementary information, see **pages 6-7**.

Reviewer #2

Comments to the Author:

In this work, authors prepared a range of mesoporous metal oxides with a method termed SPEA utilizing solvent-pair surfactant concept. This work is extensively conducted with DFT calculations, molecular simulations, experimental analysis and characterizations, and discussion is well-written. It can be considered towards acceptance after addressing the following comments.

Response: We thank the reviewer for the positive comments.

1. Figures must be discussed in order. (e.g., Fig. 1c is discussed before Fig. 1a and b,

and many other misordering of the figure number and alphabets in the manuscript).

Response: We thank the reviewer's careful reviewing and good suggestion. We reorganized the referenced order of figures for the main text and supplementary information. The referenced figures were adjusted and highlighted in the revised manuscript.

2. The demonstrated method (SPEA) seems quite similar to the existing methods utilizing soft-templates (e.g., EISA). Further discussion on different types of existing similar synthetic methods should be made, and the significance of the current SPEA method should be extensively highlighted.

Response: We thank the reviewer's good suggestion. Actually, our approach is a branch of the EISA method. The major difference between our approach and conventional soft-template method is the indirect interaction we proposed. Besides, our approach is more flexible to assemble sub-nanoscale units like POMs to precisely synthesize porous materials with rich compositions and structures. In the revised manuscript, we included a more detailed description of the methods for synthesizing mesoporous materials and the existing methods for the co-assembly of copolymers and POMs: As a family of nanostructured materials, ordered mesoporous materials, especially transition metal oxides with rich components and attractive properties have attracted ever-growing attention due to their high specific surface areas, uniform pores and adjustable morphologies and thus have a great potential in catalysis, sensing, energy storage and conversion^{11,12}. Until now, hard-template, soft-template and self-template methods were developed to synthesize mesoporous materials¹³⁻¹⁶. Among them, the soft-template method is of special concern due to its flexibility, tunability and simplicity in constructing mesostructured materials. In conventional soft-template synthesis of mesoporous materials, small surfactants or amphiphilic block copolymers can co-assemble with precursors through direct organic-organic or organic-inorganic co-assembly, wherein the host organic structure-directing agents directly co-assemble with guest precursors (e.g., silicate or aluminosilicate, synthetic resol, molecular metal salts) through intermolecular interactions, such as covalent interactions, hydrogen bonding,

\$\pi\$ - \$\pi\$ stacking and Coulombic interactions¹⁷⁻¹⁹.

Polyoxometalates (POMs) are a family of nanosized metal-oxo clusters with protons (denoted as acidic POMs) or ammonium and alkali metals (denoted as non-acidic POMs) as counter-cations. Because acidic POMs are compatible in many assembly processes using polar organic solvents, the assembly of acidic POMs and copolymers has recently emerged as an alternative route to synthesize mesoporous metal oxides (mMOs) based on the Coulombic attraction directly between heteropolyacid anions and protonated copolymer^{21,22}. However, this direct Coulombic interaction is unfavorable for the construction of mesostructured materials in a wide synthesis condition. To overcome these limitations and create mMOs with rich compositions and pore structures, it is of great importance and interest to explore a general and versatile copolymers templating route suitable for various POM precursors (see pages 2-3).

3. Although DFT calculation-based discussion is present, experimental discussions on the effect of binary systems with other organic compounds should be added.

Response: We thank the reviewer's good suggestion. We have included the experimental details about the solubility of ammonia metatungstate in binary systems with 11 kinds of organic compounds, see **pages 8-9** of the revised supplementary information.

4. 'Columbic attraction' should be revised to 'Coulombic attraction'

Response: We thank the reviewer's careful reviewing. We have corrected it throughout the revised manuscript.

5. Figure resolution must be improved, especially with the schematics.

Response: We thank the reviewer's good suggestion. We have improved the figure resolution in the revised manuscript. We will also provide vectographs to avoid distortion during uploading to the website.

6. Fig. 2 i and j seem to demonstrate cylindrical mesochannels rather than mesopores. Their TEM images should be supplied to confirm the nature of their porosity.

Response: We thank the reviewer's careful reviewing and good suggestion. We have supplied the TEM images of $mK_2Mo_3O_{10}$ and $mLiV_3O_8$ in **Fig. S12**. These two kinds of materials don't have ordered spherical mesopores like others, instead of exhibiting wormlike pores. As shown in **Fig. S7i and j**, TEM images show that the precursors ($K_3Mo_6O_{19}$ and $Li_4V_{10}O_{28}$) have low solubility in the system and exhibit as small particles. These particles cannot effectively fill the gap between PEO-*b*-PS spherical micelles. Therefore, the pore walls will partially collapse and induce the disorder transition from spherical pores to wormlike pores. In the revised manuscript, we mention that "the mesoporous $K_2Mo_3O_{10}$ and LiV_3O_8 materials possess short-range order due to the limited solubility of their precursors and the fast crystallization rate", see page 9, line 10.

Figure R6. TEM images of (a) $mK_2Mo_3O_{10}$ and (b) $mLiV_3O_8$. Mesoporous $K_2Mo_3O_{10}$ and LiV_3O_8 materials possess short-range order and wormlike mesopores.

7. Fig. 3 should be revised to what is relevant to and covered in this study (as far as I understand, this study investigated DMF and PEO-*b*-PS only among the presented organic solvents and ABCs).

Response: We thank the reviewer's good suggestion. All the contents in Fig.3 were covered in this study. The co-assembly of PEO-*b*-PS and AMT in different aqueous binary solvents was shown in **Fig. S2**. The co-assembly of AMT and different copolymers in DMF/H₂O was shown in **Fig. S24**. To avoid misunderstanding of the content, we added the description in figure notes of Fig. 3: All the contents are covered

in this study.

Figure R7. SEM images of N-WO₃ mesostructures synthesized using AMT and PEO₁₁₄-*b*-PS₂₀₀ in different aqueous binary solvent. (a) DMA/H₂O. (b) N, N-dimethylpropanamide/H₂O. (c) Pyridine/H₂O. (d) N-methylpyrrolidone/H₂O. (e) DMSO/H₂O. (f) N, N-diethylformamide/H₂O.

Figure R8. SEM images of mN-WO₃ synthesized by the co-assembly of AMT and different copolymers. (a) P4VP-*b*-PS, (b) PB-*b*-PEO, (c) PEO-*b*-PMMA and (d) PAA-*b*-PS.

8. *As this work mainly focuses on the novel synthesis of porous materials, it would benefit a broader readership to introduce porous materials prepared with other types of methods (e.g., Nature protocols 17 (12), 2990-3027; Chemical Science 13 (36), 10836-10845; Angewandte Chemie International Edition 60 (51), 26528-26534).*

Response: We thank the reviewer's good suggestion. We have included the related description in the revised manuscript, as shown in the response of Q2.

Reviewer #3

Comments to the Author:

The work by Xie et al. reports a facile and flexible "solvent-pair surfactants" enabled assembly strategy to synthesize a series of mesoporous metal oxides based on co-assembly of POM clusters and block copolymers. The developed method exhibits unique advantages over conventional EISA process, including the interesting process involving the smart and indirect assembly of polyoxometalates with block copolymers, as it exhibits greater flexibility in terms of compositions and synthesis conditions. The authors also offer new conceptual understanding for the underlying principles about dissolution and assembly process of metal-oxo clusters in binary solvents. These results are interesting and will be great help to materials scientists and chemists working in related fields. The as-designed materials show great application potential in catalysis and chemical sensing. From this point of view, I believe this paper can be published on Nature Communications after minor revisions.

Response: We thank the reviewer for the positive comments.

1. *In page 2, line 1, "uniform mesopores and stable framework was obtained and exhibits great application potentials such as gas sensing", the usage of tenses should be unified.*

Response: We thank the reviewer for the careful reviewing. We corrected the related description in the revised manuscript: nitrogen-doped mesoporous \$\epsilon\$ -WO₃ with high

specific surface area, uniform mesopores and stable framework is obtained and exhibits great application potentials such as gas sensing. See **page 2, line 1**.

2. *Figure 2 inside notes, “calcination” is incorrectly used.*

Response: We thank the reviewer for the careful reviewing. We have corrected it in the revised manuscript. See **page 25, line 5**.

3. *The scale bars in Figure 2d, g, j and m are missing.*

Response: We thank the reviewer for the careful reviewing. We have included the scale bars in the revised manuscript.

4. *In Figure S10d-g, the baseline keeps rising as the increase of temperature after the thermal degradation stage. There is also an abnormal fluctuation in Figure S10k, 250-300 °C stage. The authors should explain these data, or repeat the TG tests.*

Response: We thank the reviewer for the careful reviewing and good suggestion. We repeated the TG tests and the new data was shown in **Fig. S9**.

Figure R9. TG and DTG curves of POM precursors.

5. TG data of the PEO-*b*-PS/AMT should be given to see the decomposition behaviors of copolymer templates.

Response: We thank the reviewer for the good suggestion. We added the TG data of PEO-*b*-PS/AMT composite in **Fig. S10**. In the revised supplementary information, we mentioned this point by adding: The decomposition temperature of PEO-*b*-PS/(NH₄)₆H₂W₁₂O₄₀ is close to (NH₄)₆H₂W₁₂O₄₀, indicating that the existence of copolymers has no significant effect on the decomposition of POMs (see page 23).

Figure R10. TG and DTG curves of PEO-*b*-PS/(NH₄)₆H₂W₁₂O₄₀ composite.

6. How about the morphology of the mesoporous nitrogen-doped WO₃ after the gas sensing tests, please check if the ordered structure is retained well during the high temperature test processes.

Response: We thank the reviewer for the good suggestion. In the revised supplementary information, we revised the content by adding: After gas sensing tests and storing for one year, the mesostructures of mN-WO₃ deposited on alumina tubes can be basically retained and only accompanied by slight collapse of pore walls, indicating the good structure stability of mN-WO₃ (Fig. S44) (see page 51).

Figure R11. (a, c) SEM images of mN-WO₃ deposited on alumina tubes before gas sensing tests. (b, d) Same bath of sensors after long-term gas sensing tests and storing for one year.

7. *Why the gas sensing performances of mesoporous WO₃ prepared by conventional EISA method is inferior to mesoporous nitrogen-doped WO₃? The authors should give more discussions about the comparison of these two kinds of mesoporous materials.*

Response: We thank the reviewer for the good suggestion. We modified our description and added a more detailed discussion in the revised supplementary information: For comparison, the gas-sensing performance of aforementioned mWO₃ was investigated, while its response rate is much slower and response value is much lower than mN-WO₃ (Fig. S40g). The mWO₃ with polycrystalline walls has large amounts of grain boundaries, which may hinder the rapid migration of electrons. By contrast, the iso-oriented crystalline walls of mN-WO₃ facilitate the carriers' migration and lead to a faster response rate. Besides, the nitrogen doping enhances the polarity of mN-WO₃, which causes the relatively strong interactions between acetone and mN-WO₃, and thus further accelerate the response and lead to higher response value due to the enhanced acetone adsorption and charge transfer (see page 50, Supplementary Information).

8. *In Table S7, the comparative literature needs to be updated.*

Response: We thank the reviewer for the good suggestion. We have updated the comparative literature in the revised supplementary information, see page 56, Table S6.

9. *For the multicomponent co-assembly process, how did the author guarantee that the noble metal nanoparticles loaded on the surface of mesopores, instead of embedding inside the WO₃ matrix. If it is the latter case, the integration of noble metal and WO₃ doesn't seem to make much sense. The authors should give more discussions on it.*

Response: We thank the reviewer for the careful reviewing. In this method, Pt precursors (H₂PtCl₆) are intensely incorporated into the hydrophilic domains of BCPs due to Coulombic interactions between protonated PEO segment and PtCl₆²⁻ anions. While AMT only spreads around and fulfills the space between PEO-*b*-PS micelles due to the weak indirect interactions. After the thermal treatment, Pt precursors which are

closer to the inside of micelles will transform into Pt nanoparticles embedded on the surface of the WO_3 matrix. We included the scheme description and HRTEM images in **Fig. S56**, Supplementary Information.

Figure R12. The co-assembly of AMT and H_2PtCl_6 . (a) Scheme illustration of the synthesis of Pt-loaded mN- WO_3 . (b) Energy dispersive spectrum and (c) XRD patterns of Pt-loaded mN- WO_3 (4.6 mol% Pt, green curve) and N- WO_3 (blue curve). (d) SEM image, (e-i) TEM images and (j) HAADF-STEM image of Pt-loaded mN- WO_3 . (k-n) Element mapping shows the uniform distribution of W, Pt, O and N throughout the Pt-loaded mN- WO_3 (4.6 mol% Pt), indicative of a homogenous Pt loading.

10. In Figure 5, left bottom, some elements are unmatched. The authors should explain the meaning of these matched and unmatched element groups.

Response: We thank the reviewer for the careful reviewing and good suggestion. We

added a description in the figure notes of Fig. 5: The matched element groups at the left bottom show the combination of elements covered in this study and unmatched groups show other possible combinations.

Reviewer #4

Comments to the Author:

The article reports the well ordered synthesis of a several metal oxides based on POMs assembled with diblock polymers (BCP) and DMF/water mixtures. While POM/BCP assemblies are not novel alone (DOI: 10.1039/C3TA10333A), the main claim of this manuscript is the “solvent pair surfactants enabled assembly” concept which was suggested to operate without direct interaction of the POM with the BCP. Such non-direct interactions have been widely reported with surfactant based assemblies using counter ion mediation (DOI: 10.1038/368317a0).

Response: We thank the reviewer’s careful reviewing. The co-assembly of POMs and BCPs has recently emerged as an alternative route to synthesize mMOs. However, the existing researches only cover the assembly of Keggin-type POMs with protons as counter-cations (i.e., acidic POMs or heteropolyacid). Through these methods, POM units are selectively incorporated into the hydrophilic domains of BCPs due to Coulombic interactions between protonated nitrogen or oxygen-containing hydrophilic domains and metal-oxo anions. The direct and strong Coulombic interaction has great influences on the co-assembly system. The proton transfer from heteropolyacid to nitrogen-containing hydrophilic domains results in instant precipitation of the complex, similar to the alkaloid precipitant reaction process. To avoid this unfavorable effect, the hydrophobic domains need to maintain a high degree of polymerization to ensure the dissolution of the complex, and thus the type of mesophase tends to be shown as cylindrical pores controlled by the Florry-Huggin parameter (DOI: 10.1021/ja304073t; 10.1002/anie.201206183; 10.1039/c3ta10333a). Other works adopt similar ideas and obtain some novel morphologies such as 3D cross-stacked nanowires (DOI: 10.1038/s41563-019-0542-x). These results are interesting, but in the meantime,

lacking flexibility and controllability for tailored synthesis of mesoporous materials.

Besides, most of the metal-oxo clusters are non-acidic POMs with ammonium or alkali metals as counter-cations which cannot provide protons. The pH is a main factor in POM speciation because the structure of common metal-oxo clusters only keeps stable over a narrow pH range (DOI: 10.1126/sciadv.adi0814). It must be very careful to add exogenous protons which may destroy the structure of metal-oxo clusters or hinder the incorporation of intrinsic counter-cations into obtained mesostructures. Therefore, it is of great importance and interest to explore a general and versatile BCPs templating route suitable for various POM precursors.

The SPEA method we developed enables the general assembly of POMs as well as their counter-cations. Because the non-acidic POM precursors don't have direct and strong interactions with BCP templates, the morphologies of BCP aggregates can be well tuned according to the classical Flory-Huggin parameters without the interference of host-guest interactions between copolymers and POMs. After the calcination, mesoporous metal oxides with different pore sizes and structures can be obtained.

The classical electrostatic interaction powered assembly process is controlled by electrostatic complementarity between the inorganic ions in solution and the charged surfactant head groups, and inorganic counterions when these charges both have the same sign ($S^+X^-I^+$ and $S^-X^+I^-$). The inorganic counterions come from either surfactants or inorganic precursors which are essential parts of the assembly units (DOI: 10.1038/368317a0). There are also some previous reports that use bridge ligands to enhance the interactions between copolymer templates and inorganic precursors by means of hydrogen bonding or coordination interactions (DOI: 10.1002/anie.201303353; 10.1021/jacs.6b11411). Although these inorganic counterions or bridge ligands may be removed by calcination or washing, they are indeed parts of the assembly units and incorporated into the obtained as-made composites. From this viewpoint, these reports are different from the indirect interactions we discussed. In our work, the DMF- nH_2O complexes just behave as carriers to allocate non-acidic POMs at the hydrophilic region of micelles and will evaporate after fulfilling their mission as solvents.

In the revised manuscript, we included a detailed description of the existing method for the co-assembly of POMs and copolymers: Polyoxometalates (POMs) are a family of nanosized metal-oxo clusters with protons (denoted as acidic POMs) or ammonium and alkali metals (denoted as non-acidic POMs) as counter-cations. Because acidic POMs are compatible in many assembly processes using polar organic solvents, the assembly of acidic POMs and copolymers has recently emerged as an alternative route to synthesize mesoporous metal oxides (mMOs) based on the Coulombic attraction directly between heteropolyacid anions and protonated copolymer^{21,22}. However, this direct Coulombic interaction is unfavorable for the construction of mesostructured materials in a wide synthesis condition. To overcome these limitations and create mMOs with rich compositions and pore structures, it is of great importance and interest to explore a general and versatile copolymers templating route suitable for various POM precursors (see page 3).

The following issues were of significant concern:

1) It seems that the reported data are consistent with traditional polymer structure directing agent approaches (DOI: 10.1557/s43578-021-00421-0) where the DMF/water mixtures are simply a solvent-system optimization to enable solubility of all species. How can the claimed “SPEA” mechanism be distinguished from standard coassembly behaviors? Would not one anticipate hydrophilic polymers to naturally interact with the surfaces of POMS so long as they were well solubilized by the solvent mixture? From this perspective, the main claimed mechanism is not supported.

Response: We thank the reviewer’s careful reviewing. It is not easy to simply dissolve the assembly units in a homogenous system and induce them to assemble into ordered structures. The “SPEA” mechanism is mainly based on the microscopic homogeneity of binary solvent. Taking the DMF-H₂O system as an example, microscopic homogeneity at the molecular level occurs in the case of low H₂O content (< 20 vol%). When the water content exceeds 20 vol%, the DMF-nH₂O complexes are not dominant H₂O species and replaced by H₂O clusters. In this case, it is impossible to obtain an

ordered assembled mesostructure even if all the components are well solubilized by the solvent mixture, as shown in Fig. S28.

In conventional organic-inorganic co-assembly process, specific host-guest interactions are usually given to reveal the assembly mechanisms, including hydrogen bonding, covalent bonding, π - π stacking, coordination and electrostatic interactions. Few literatures directly describe the driving force of assembly in terms of hydrophilic interaction. For the co-assembly of POMs and BCPs, all the previous reports propose electrostatic interaction as the mechanism (DOI: 10.1039/c9ce00228f).

Hydrophilic interaction is usually used to describe macroscale or nanoscale interactions and it has more specific forms at the molecular scale (e.g., hydrogen bonding or dipole-dipole interaction). The “SPEA” process by means of hydrophilic interaction in general terms and we try to give a clearer understanding of the underlying mechanism. The surface of metal-oxo clusters is covered by oxygen atoms and the functional groups of electroneutral PEO segments are ether bonds. Although PEO and POMs are both hydrophilic units, they cannot attract each other because they are both electron donors and don't have electron acceptors. The counter-cations of non-acidic POMs are ammonium or alkali metal ions. These ions don't have empty “d” or “f” orbitals. None of the hydrogen bonding, coordination, or electrostatic interactions could interact electroneutral PEO segments with these ions. Therefore, there are certainly some carriers (i.e., DMF-nH₂O complexes) that enable the selective location of metal-oxo clusters and their counter-cations among BCP micelles.

2) The generalizability was not particularly compelling since all of the BCPs used had generic hydrophilic blocks (PEO, PAA, or P4VP), each of which have previously been demonstrated with BCP SDAs using sol-gel routes. I.e. a suitable solvent system for POMs would have been anticipated to yield such assemblies with any generic amphiphilic SDA. This returns to the point of the “SPEA” mechanism not being directly supported. Here the results are both understandable and expected from standard hydrophilic interactions of POMs with SDAs.

Response: We thank the reviewer's careful reviewing. For the construction of

nanoarchitectures, hydrophilic blocks with different properties are usually used to synthesize specific kinds of materials with the purpose of providing a more suitable driving force for assembly. (DOI: 10.1038/NNANO.2013.85; 10.1002/adma.202100820). Although all these copolymers were used to prepare some nanoarchitectures (e.g., mesoporous SiO₂) which are easy to synthesize, the experiment parameters (i.e., pH or catalysts) and interaction mechanisms of them have slight differences (DOI: 10.1021/ja1070653; 10.1021/ja207525e; 10.1002/adma.202100820).

As discussed above, we think the hydrophilic interaction has more specific forms at the molecular scale. In this study, direct interactions between copolymers and POMs are not required due to the existence of DMF-nH₂O complexes as carriers. Therefore, the assembly of non-acidic POMs is not disturbed even if copolymers with different kinds of charges are used which may produce slight repulsion or attraction with metal-oxo clusters. We chose these three kinds of representative copolymers to verify the generality of our methods. Our synthesis can be easily repeated and reproducible under the same experiment conditions without the restriction of copolymer species.

3) The authors claim that iso-oriented walls cause high temperature stability. That claim would need to be supported with evidence such as an otherwise identical sample with non-oriented walls which exhibited less temperature stability. Rather than a real claim, this appears to be a random observation where others have also seem iso-oriented samples previously from SDAs (DOI: 10.1038/NMAT2612).

Response: We thank the reviewer's good suggestion. We synthesized mesoporous WO₃ by using WCl₆ as precursor following a conventional evaporation-induced assembly process as reported by previous works (DOI: 10.1021/jacs.7b04221; 10.1002/adfm.201705268). The molecular weight of PEO-*b*-PS (PEO₁₁₄-*b*-PS₂₀₀) templates and thermal treatment conditions kept consistent with this work. As revealed by SEM and TEM images (**Fig. S32**), the obtained mWO₃ has ordered spherical mesopores and polycrystalline pore walls, which is consistent with previous reports. After the thermal treatment at 700 °C in N₂ for 0.5 h, the pore walls of mWO₃ underwent collapse to form scattered particles due to the fusion of grains, which is proved by SEM,

TEM and SAXS characterizations (**Fig. S34, S35**). By contrast, the mN-WO₃ prepared with POM precursors maintained the ordered structure.

We found that iso-oriented crystalline units are a property of POM-derived metal oxide nanoarchitectures. Similar properties were found in the HRTEM and SAED data of previous work (DOI: 10.1038/s41563-019-0542-x). Both Keggin-type POMs and orthorhombic WO₃ are constructed by W-O octahedrons. The similarity of the crystalline structure of precursors and products may contribute to the formation of iso-oriented crystalline walls.

Reseachers have found iso-oriented samples previously. Iso-oriented mesoporous MoO₃ and Nb₂O₅ films were obtained by using the “soft epitaxy” method (DOI: 10.1002/adma.200600154; 10.1038/NMAT2612). Radially oriented single-crystal-like mesoporous TiO₂ microspheres were obtained by using evaporation induced aggregation assembly method (DOI: 10.1126/sciadv.1500166). Mesoporous TiO₂ single crystals were prepared by using a hydrothermal approach (DOI: 10.1038/nature11936). All these methods are effective while also exhibiting some limitations. Our work provides an alternative synthesis concept in efficiently enriching the methodology in this field.

Figure R13. (a) Typical SEM and (b) TEM images with a selected-area electron

diffraction pattern of $m\text{WO}_3$ by using WCl_6 as precursor. (d). High-resolution TEM image of the selected area in (b). The sample was obtained after the thermal treatment at 500 °C in N_2 for 0.5 h and then at 400 °C for 0.5 h in air.

Figure R14. (a) SEM and (c) TEM images of $m\text{N-WO}_3$ synthesized by using AMT as tungsten precursor after the thermal treatment at 700 °C in N_2 for 0.5 h. (b) SEM and (d) TEM images of $m\text{WO}_3$ synthesized by using WCl_6 as tungsten precursor after the thermal treatment at 700 °C in N_2 for 0.5 h. The scale bar is 100 nm.

Figure R15. Small angle X-ray scattering spectrum of the $m\text{N-WO}_3$ and $m\text{WO}_3$ after the thermal treatment at 700 °C in N_2 for 0.5 h.

Additionally, the following aspects were of minor concern:

4) *The authors claimed tailored pore size, structure, and pore-wall environment. Were those really tailored by choice, if so what were the guiding principles? Rather it appears that these were a collection of arbitrary observations expected with “dynamic micelle templates”.*

Response: We thank the reviewer’s careful reviewing. PEO-*b*-PS micelles used in our work are persistent micelles as they have sufficient barriers toward chain exchange, especially at aqueous solution (DOI: 10.1039/d1ma00146a). In our study, the AB copolymers assemble with non-acidic POMs through indirect interaction, thus the structure of copolymer aggregates is mainly determined by critical packing factor $p = V/a_0l_c$. V and l_c represent the volume and length of the hydrophobic segment, respectively. a_0 expresses the molecular area of the hydrophilic block. The increase of PS block leads to an increase in V/l_c and consequently the p value also increases. Given the fixed solvent composition of DMF/H₂O (9:1 v/v), the topological transition from spherical (PEO₁₁₄-*b*-PS₁₀₃₋₂₅₀), cylindrical (PEO₁₁₄-*b*-PS₂₆₅) to lamellar (PEO₁₁₄-*b*-PS₂₇₅) micelles was evolved as the increase of PS block length. Before the topological transition from spherical to cylindrical micelles, the increase of PS block length only enlarges the pore size because the hydrophobic segment usually determines the pore size and the hydrophilic segment determines the wall thickness (see **Fig. S29, S30**).

For the co-assembly of POMs and molecular species to synthesize heteroatom-doped or noble metal-loaded mesoporous materials, the guest materials (heteroatoms or noble metals) interact with copolymers by means of coordination or electrostatic interactions. The additive amount of guest materials is quite low, thus providing more options. We chose metal chloride, acetate or nitrate as precursors which hardly hydrolyze and the anion can be easily removed during the evaporation or calcination process. For the construction of mesoporous metal oxide composites, all the precursors are POMs because of the high content of both host and guest materials. The experimental details were added to **pages 6-7, Supplementary Information**.

5) *The authors claimed that traditional EISA approaches do not enable tunable*

structures which is quite incorrect (DOI: 10.1021/cm011209u)

Response: We thank the reviewer's carefully reviewing. In the revised manuscript, we modified our description and discussion about this point: these assembly processes lack the flexibility in constructing periodic nanoarchitectures with rich compositions in a simple system. (See **page 3, line 4**)

6) The authors seem confused about polymer nomenclature and routinely call diblock polymers "ABCs." That term is reserved most properly for triblock terpolymers whereas all of the polymers used here are more accurately called AB polymers.

Response: We thank the reviewer's good suggestion. We have changed all of the "ABCs" to "AB copolymers".

7) The authors claim that "different types of direct host-guest interactions may interfere with each other and lead to macroscopic phase separation." The authors should be more clear in what they are trying to say as these words as-written appear self-contradictory.

Response: We thank the reviewer's carefully reviewing. We want to express that multiple precursors simultaneously assembling with amphiphilic copolymers may interference with each other, which was reported by previous work (DOI: 10.1021/acscentsci.1c00912). Therefore, it needs to take measures to bridge or separate different kinds of precursors in the system (DOI: 10.1002/sml.201904240). For example, for design and synthesis of Pt-loaded mesoporous WO₃ (DOI: doi.org/10.1002/adfm.201705268; 10.1021/acscentsci.1c00912), hydrophobic platinum precursors and hydrophilic tungsten precursors were used to rationally allocate the two kinds of precursors at the region of hydrophobic and hydrophilic segments of copolymers, respectively. Considering that the expression of this sentence may lead to ambiguity, we delete this description in the revised manuscript.

Reviewers' Comments:

Reviewer #1:

Remarks to the Author:

I want to express my appreciation for the effective revisions you've made to the paper in response to the requests. Thank you for your hard work and for ensuring that the paper now meets the desired standards.

Reviewer #2:

Remarks to the Author:

As the revision has been done at a satisfactory level, I recommend acceptance of this manuscript.

Reviewer #3:

Remarks to the Author:

The manuscript has been well revised, and it can be accepted.

Reviewer #4:

Remarks to the Author:

The revised manuscript has improved significantly. The novelty of the water/DMF concept is now better explained as a conceptual extension of the reported indirect ionic-interactions for surfactant structure directing agents.

The central claim of the manuscript needs direct evidence. Beyond computation, is there experimental evidence for the proposed mediation of POM-water/DMF-polymer? Could it be that the polymer directly interacts with the POM surface once it is dissolved, similar to e.g. THF coordination of inorganic salts (analogous to PEO ether interaction)? Perhaps some IR or multi-dimensional NMR experiments could indicate the specific species that is interacting with PEO? Lacking such evidence, these remarkable observations may simply be related to solvation conditions.

As a side note, the authors claim that "Given the fixed solvent composition of DMF/H₂O (9:1 v/v), the topological transition from spherical (PEO114-b-PS103-250), cylindrical (PEO114-b-PS265) to lamellar (PEO114-b-PS275) micelles was evolved as the increase of PS block length." If this is the case, then they should be able to detect cylindrical and lamellar micelles in the starting solutions. Without suitable evidence, the authors should not speculate about the underlying thermodynamics and kinetics. I think that more likely, the large amount of DMF present plasticizes the PS and leads to dynamic reorganization of spherical micelles into bulk phases of block polymers. Evidence is only needed when such claims are made. Since this is a side point of the manuscript, the authors are suggested to simply bring the claims in line with evidence.

Response Letter

Manuscript ID: NCOMMS-23-25577-A

Title: “Solvent-Pair Surfactants” Enabled Assembly of Clusters and Copolymers towards Programmed Mesoporous Metal Oxides

General response: We sincerely thank the editor and all the reviewers for their valuable and positive comments. The reviewer comments are in italic font below, and specific concerns are numbered. Our response is given in normal blue font, and all the changes/additions in the revised manuscript are highlighted in yellow.

Response to Review’s Comments

Reviewer #4

Comments to the Author:

The revised manuscript has improved significantly. The novelty of the water/DMF concept is now better explained as a conceptual extension of the reported indirect ionic-interactions for surfactant structure directing agents.

Response: We thank the reviewer’s help for improving the quality of our manuscript.

The central claim of the manuscript needs direct evidence. Beyond computation, is there experimental evidence for the proposed mediation of POM-water/DMF-polymer? Could it be that the polymer directly interacts with the POM surface once it is dissolved, similar to e.g. THF coordination of inorganic salts (analogous to PEO ether interaction)? Perhaps some IR or multi-dimensional NMR experiments could indicate the specific species that is interacting with PEO? Lacking such evidence, these remarkable observations may simply be related to solvation conditions.

Response: We thank the reviewer’s careful reviewing and good suggestion. In the revised manuscript, further characterization was performed using Fourier transform infrared spectroscopy (FTIR) and the results show that the spectrum of PEO-*b*-PS/(NH₄)₆H₂W₁₂O₄₀ composite is a simple additive combination of the spectra of PEO-*b*-PS and (NH₄)₆H₂W₁₂O₄₀ (Figure R1). New characteristic peaks and significant shift of peak position were not found, implying that no new bonds or strong intermolecular

interactions were formed (e.g., hydrogen bond and coordination bond). Yamauchi and Liu et al. used UV-Vis to verify the existence of intermolecular interactions (DOI: 10.1039/c7mh00586e; 10.1002/advs.202301918), because new peaks can appear once the molecules interact with each other through electrostatic, coordination or dipole-dipole interactions. UV-vis spectra revealed that the mixing of PEO-*b*-PS and (NH₄)₆H₂W₁₂O₄₀ in DMF/H₂O system did not produce new peaks, providing additional evidence that there are no strong intermolecular interactions (Figure R2a). For comparison, we prepared the mixture of P4VP-*b*-PS and heteropolyacid (H₃PW₁₂O₄₈) in DMF/H₂O, and it was found that the metal-oxo anions interact with protonated copolymer through electrostatic interaction, forming new peak at ~ 500 cm⁻¹ assigned to P4VP-*b*-PS/H₃PW₁₂O₄₀ hybrid complex (Figure R2b). These results clearly confirmed that PEO-*b*-PS copolymers has no direct interactions with (NH₄)₆H₂W₁₂O₄₀ in our synthesis system.

Furthermore, proton NMR was used to explore the microenvironment of the PEO segments after introduction of H₂O. As shown in Figure R3, compared to spectrum (a) for DMF/H₂O/PEO-*b*-PS, the peak of the H₂O protons in spectrum (b) for DMF/H₂O became narrowed in width and its position shifted slightly from 4.12 to 4.13 ppm. Such a phenomenon can be explained as follows. The line widths of the H₂O proton NMR spectra are directly relevant to the distribution of H₂O in the solution. In this study, H₂O as a small molecule shows a narrow resonance peak due to its high mobility, while PEO-*b*-PS copolymers possessing long chains with high molecular weight and low mobility display relatively broad NMR peaks. Thus, in this study, because H₂O molecules are selectively accumulated around or interact with PEO chains, their mobility decreases and the NMR peak is broadened.

Based on these characterizations, combining with theoretical calculations and simulations, we conclude that non-acidic POMs mainly interact with copolymers through the mediation of DMF/H₂O pairs. We supplied these results in Fig. S6-8 and included the related description in the revised manuscript: Fourier transform infrared (FTIR) and UV-vis absorption spectra confirm that POMs have no direct strong intermolecular interactions with PEO-*b*-PS (Supplementary Fig. 6 and 7). Proton

nuclear magnetic resonance spectra reveal that the full width at half maximum of the water proton peak becomes larger in PEO-*b*-PS/DMF/H₂O solution compared with pure DMF/H₂O solution, indicating the accumulation of water preferentially around the PEO blocks (Supplementary Fig. 8) (See page 8).

We noticed that DMF and PEO could interact with some inorganic salts through coordination or hydrogen bonds. However, they usually interact with cations, especially those having empty *d* or *f* orbitals. Non-acidic POMs are quite special. The metal-oxo clusters are covered by oxygen which cannot directly interact with the ether bonds of PEO. On the other hand, the counter-cations are alkali metal or ammonium ions, and the interactions between these ions and PEO in DMF solution are relatively weak. These counter-cations, especially ammonium ions, cannot drive the assembly of metal-oxo clusters and PEO-*b*-PS copolymers. However, the acidic POMs have good solubility in organic solvent and show flexibility in interacting with other molecules by virtue of their protons, and this is why the assembly of acidic POMs and amphiphilic molecules were widely reported.

Figure R1. FTIR spectra of PEO-*b*-PS, (NH₄)₆H₂W₁₂O₄₀·xH₂O and PEO-*b*-PS/(NH₄)₆H₂W₁₂O₄₀ composites after thermal treatment at 100 °C to remove the solvents.

Figure R2. UV-vis absorption spectra of (a) PEO-*b*-PS, $(\text{NH}_4)_6\text{H}_2\text{W}_{12}\text{O}_{40}$, PEO-*b*-PS/ $(\text{NH}_4)_6\text{H}_2\text{W}_{12}\text{O}_{40}$ and (b) P4VP-*b*-PS, $\text{H}_3\text{PW}_{12}\text{O}_{40}$, P4VP-*b*-PS/ $\text{H}_3\text{PW}_{12}\text{O}_{40}$ in DMF/ H_2O (9:1 v/v).

Figure R3. Proton NMR spectra of 9(DMF-d7)-1 H_2O binary solution with (a) or without (b) 0.01 g/ml PEO-*b*-PS.

As a side note, the authors claim that “Given the fixed solvent composition of DMF/ H_2O (9:1 v/v), the topological transition from spherical (PEO₁₁₄-*b*-PS₁₀₃₋₂₅₀), cylindrical (PEO₁₁₄-*b*-PS₂₆₅) to lamellar (PEO₁₁₄-*b*-PS₂₇₅) micelles was evolved as the increase of PS block length.” If this is the case, then they should be able to detect cylindrical and lamellar micelles in the starting solutions. Without suitable evidence, the authors should not speculate about the underlying thermodynamics and kinetics. I think that more likely, the large amount of DMF present plasticizes the PS and leads to dynamic reorganization of spherical micelles into bulk phases of block polymers.

Evidence is only needed when such claims are made. Since this is a side point of the manuscript, the authors are suggested to simply bring the claims in line with evidence.

Response: We thank the reviewer's careful reviewing and good suggestion. We directly dropped as-prepared diluted PEO-*b*-PS/POMs co-assembly system onto carbon-coated copper grids for TEM characterization. TEM images (Figure R4a and d) confirm the existence of cylindrical and lamellar micelles even in the diluted solutions, which is consistent with SEM observations (Figure R4b and e). The cylindrical micelles observed by TEM (Figure R4a) are shorter than the cylindrical pores of final products (Figure R4c), which implies the further growth of cylindrical micelles with the increase of the concentration of copolymers. From this point of view, the large amount of DMF can indeed plasticize the PS and make the micelles more flexible to growth, but may not change the topologies. We supplied the relative images in Fig. S33.

Figure R4. (a) TEM and (b) SEM images of as-made PEO₁₁₄-*b*-PS₂₆₅/AMT composites. (c) SEM image of mN-WO₃ synthesized using PEO₁₁₄-*b*-PS₂₆₅ as template. (d and inset) TEM and (e) SEM images of as-made PEO₁₁₄-*b*-PS₂₇₅/AMT composites. (f) SEM image of mN-WO₃ synthesized using PEO₁₁₄-*b*-PS₂₇₅ as template. The scale bars are 200 nm.

Reviewers' Comments:

Reviewer #4:

Remarks to the Author:

The new UV-vis data could be compelling. The authors show a new peak at 500 nm, suggestive of a new intermolecular interaction for P4VP-b-PS/H3PW12O40 in DMF/H₂O. What do the UV-vis patterns look like for the same polymer and POM combination in pure water and pure DMF, separately? In other words, does this peak at 500 nm require the specific DMF/water mixtures claimed for the SPEA mechanism or does it form with any combination of good solvents? Does this signal come and go with the specific DMF/water ratios indicated in the manuscript?

Casting these DMF/water solutions onto a TEM grid is not a method to validate the solution phase. The water evaporates first, the DMF will plasticize all the blocks and perhaps enable transition to bulk phases. The suggestion remains to bring the morphology claims in line with the evidence: dynamic equilibration seems to be at play.

Small note, Figure R2a is uninterpretable with out the solvent specified for the polymer/POM combination.

The single-dimensional NMR data is vaguely suggestive, not worth much interpretation.

Response to Review's Comments

Reviewer #4

Comments to the Author:

Comment 1. The new UV-vis data could be compelling. The authors show a new peak at 500 nm, suggestive of a new intermolecular interaction for P4VP-*b*-PS/H₃PW₁₂O₄₀ in DMF/H₂O. What do the UV-vis patterns look like for the same polymer and POM combination in pure water and pure DMF, separately? In other words, does this peak at 500 nm require the specific DMF/water mixtures claimed for the SPEA mechanism or does it form with any combination of good solvents? Does this signal come and go with the specific DMF/water ratios indicated in the manuscript?

Response: We appreciate the reviewer's comments and suggestion. We are very sorry that we did not describe the experimental details for P4VP-*b*-PS/H₃PW₁₂O₄₀ assembly and our motivation clearly enough in the last response letter for previous revision, and this misled the reviewer to think that P4VP-*b*-PS/H₃PW₁₂O₄₀ system follows the same SPEA mechanism. Actually, in the last revision, to further disclose the SPEA mechanism in the case of PEO-*b*-PS/(NH₄)₆H₂W₁₂O₄₀, the P4VP-*b*-PS/H₃PW₁₂O₄₀ system was used as a comparison system which does not follow the SPEA mechanism. Here, we would like take this chance to give more descriptions about the motivation and experimental results.

In order to provide more evidence for the proposed SPEA process requiring no interactions between polymers and non-acidic POMs, we employed UV-vis spectroscopy to study whether PEO-*b*-PS has direct strong intermolecular interactions with (NH₄)₆H₂W₁₂O₄₀ or not. We performed the UV-vis analysis by mixing the PEO-*b*-PS and (NH₄)₆H₂W₁₂O₄₀ in DMF/H₂O (9:1 v/v) according to typical experimental conditions in our study. The UV-vis spectra reveal that PEO-*b*-PS/DMF/H₂O, (NH₄)₆H₂W₁₂O₄₀/DMF/H₂O, and the mixed PEO-*b*-PS/(NH₄)₆H₂W₁₂O₄₀/DMF/H₂O all show similar absorption patterns without significant peaks, implying that there are no direct strong intermolecular interactions between PEO-*b*-PS and (NH₄)₆H₂W₁₂O₄₀.

Furthermore, for comparison study, we prepared another assembly system by mixing P4VP-*b*-PS and acidic POMs (H₃PW₁₂O₄₀) in DMF/H₂O (9:1 v/v). This well-

documented acidic POMs/polymer assembly system experiences direct intermolecular interactions and exhibits a totally different UV-vis pattern (Figure R1b), compared to our proposed non-acidic POMs/PEO-*b*-PS assembly system, namely the SPEA process (Figure R1a). The UV-vis analysis reveals that, compared with P4VP-*b*-PS/DMF/H₂O and H₃PW₁₂O₄₀/DMF/H₂O, the P4VP-*b*-PS/H₃PW₁₂O₄₀/DMF/H₂O has a new absorption peak at 500 nm, due to the formation of lots of protonated pyridine groups in the P4VP segments enabling the strong electrostatic interactions between positively charged P4VP segments and PW₁₂O₄₀³⁻ anions (inset in Figure R1b).

Accordingly, in order to avoid misunderstanding and give concise descriptions, we modified the related description in the revised Supplementary Information (**see SI pages 18-19**).

Comment 2. Casting these DMF/water solutions onto a TEM grid is not a method to validate the solution phase. The water evaporates first, the DMF will plasticize all the blocks and perhaps enable transition to bulk phases. The suggestion remains to bring the morphology claims in line with the evidence: dynamic equilibration seems to be at play.

Response: We thank the reviewer for the thoughtful comment and suggestion. We agree that, in our study, there is some possibility that DMF can plasticize the non-acidic POM/PEO-*b*-PS composite and affect their assembled morphologies. According to the reviewer's suggestion, we modified our description in the revised manuscript: The pore structures can be adjusted from spherical, cylindrical to lamellar by changing the hydrophobic segment length of the AB copolymers and tuning the dynamic equilibration of the composite micelles under specific conditions (**see page 13**).

Comment 3. Small note, Figure R2a is uninterpretable without the solvent specified for the polymer/POM combination.

The single-dimensional NMR data is vaguely suggestive, not worth much interpretation.

Response: We thank the reviewer for the careful reviewing and constructive suggestion. The solvent composition for the polymer/POM combination is the same as the typical

assembly system ($V_{\text{DMF}}: V_{\text{H}_2\text{O}} = 9:1$). In the revision, we added the description in the figure note, as shown in Figure R1.

We agree that the NMR data cannot give too much information about the mechanism. Therefore, we deleted some unnecessary discussion in the revised manuscript (see page 8).

Figure R1. UV-vis absorption spectra of (a) PEO-*b*-PS, $(\text{NH}_4)_6\text{H}_2\text{W}_{12}\text{O}_{40}$, PEO-*b*-PS/ $(\text{NH}_4)_6\text{H}_2\text{W}_{12}\text{O}_{40}$ all in DMF/ H_2O (9:1 v/v) and (b) P4VP-*b*-PS, $\text{H}_3\text{PW}_{12}\text{O}_{40}$, P4VP-*b*-PS/ $\text{H}_3\text{PW}_{12}\text{O}_{40}$ all in DMF/ H_2O (9:1 v/v).

Reviewers' Comments:

Reviewer #4:

Remarks to the Author:

If I understood the authors correctly, the lack of any meaningful signals in Fig S7a is claimed to confirm the SPEA process with non-acidic POMs. The lack of a signal is not strong evidence for the existence of something specific.

Why is not data shown for a specific unique SPEA signal (absorption, FTIR, Raman, etc) from e.g. PEO-b-PS/(NH₄)₆H₂W₁₂O₄₀/DMF/H₂O, to evidence the claimed mechanism? Such data would need the corresponding control samples to validate that some specific SPEA signal only occurs with specific combinations of ingredients. The old adage is that "amazing claims require amazing evidence," where here the lack of a signal does not satisfy the burden of evidence.

This being the case, the observations here remain consistent with simply solubilizing the non-acidic POMs with a specific DMF/water mixture.

Manuscript ID: NCOMMS-23-25577-C

Response to Review's Comments

Reviewer #4

Comments to the Author:

If I understood the authors correctly, the lack of any meaningful signals in Fig S7a is claimed to confirm the SPEA process with non-acidic POMs. The lack of a signal is not strong evidence for the existence of something specific.

Why is not data shown for a specific unique SPEA signal (absorption, FTIR, Raman, *etc*) from *e.g.* PEO-*b*-PS/(NH₄)₆H₂W₁₂O₄₀/DMF/H₂O, to evidence the claimed mechanism? Such data would need the corresponding control samples to validate that some specific SPEA signal only occurs with specific combinations of ingredients. The old adage is that “amazing claims require amazing evidence,” where here the lack of a signal does not satisfy the burden of evidence.

This being the case, the observations here remain consistent with simply solubilizing the non-acidic POMs with a specific DMF/water mixture.

Response: We appreciate the reviewer's comments and constructive suggestions. For the SPEA process, we conclude that non-acidic POMs interact with PEO-*b*-PS through the mediation of DMF/H₂O pairs, as shown in **Figure R1**. According to your kind suggestion, in the revised manuscript, further characterization was performed using ¹H NMR spectroscopy to investigate the interactions between these components. PEO (M_w = 350) homopolymer was used to replace the PEO-*b*-PS during the NMR characterization because the PEO segments were considered to interact with water molecules in the SPEA process and the long PS segments can suppress the signals of the components of interest, including PEO and H₂O.

A representative ¹H NMR characterization result of SPEA process in DMF-d₇/D₂O (9:1 v/v) is shown in **Figure R2b**. As the content of PEO increases, the H_a (hydrogen of H₂O) signal shifts upfield. Meanwhile, the H_b (hydrogen of methylene in PEO) signal also shifts upfield but the amplitude is smaller. These results indicate that PEO interacts with H₂O through double hydrogen bonding in opposite directions. Into this system, we

further introduce POMs ($(\text{NH}_4)_6\text{H}_2\text{W}_{12}\text{O}_{40}$), which can induce the signal of H_a to shift downfield (Figure R2b). For comparison, we directly add POMs to DMF- d_7 / D_2O (9:1 v/v) and we found that as the content of POMs increases, the H_a signal also shifts downfield (**Figure R3**) but the amplitude is smaller than that in Figure R2b. Therefore, the obvious chemical shifts of H_a in Figure R2b can be attributed to the following two aspects. Firstly, due to the oxygen-rich surface, POMs can interact with H_2O through hydrogen bonding ($\text{O}-\text{H}\cdots\text{O}$). Secondly, the POMs- H_2O interaction can weaken the hydrogen bonding between PEO and H_2O . In other words, POMs and PEO simultaneously interact with H_2O (or $\text{DMF}\cdot n\text{H}_2\text{O}$ complexes), which implies that H_2O can behave as a medium to connect with POMs and PEO. Moreover, the noticeable broadening of the H_a signal with the POMs content indicates that POMs have relatively strong interactions with H_2O , retaining H_2O molecules tightly around the POMs.

In the ^1H - ^1H NOESY spectrum of PEO/POMs/ H_2O /DMF mixture (**Figure R4**), some weak but visible cross peaks between H_a and H_b can be clearly detected, further confirming that H_2O is spatially close to PEO. The hydrogen of $\text{H}_2\text{W}_{12}\text{O}_{40}^{6-}$ is covered by W-O octahedrons and cannot contact with other species. Therefore, the relevant cross peaks of the hydrogen of POMs are not detected.

According to the reviewer's suggestion, we further perform control experiments to validate the SPEA process. In pure DMF- d_7 , the addition of PEO and POMs does not cause any significant chemical shift, indicating that there is no obvious interaction between the components in this system (**Figure R2a**). When the excessive content of H_2O is applied (**Figure R2c**, $V_{\text{DMF-d}_7}/V_{\text{D}_2\text{O}} = 2:1$, $V_{\text{D}_2\text{O}} \sim 33\%$), the addition of POMs only causes the H_a signal to shift downfield slightly, and this shift amplitude is much smaller than that in the 9DMF- d_7 -1 D_2O system, and no obvious broadening occurs. It suggests that the interactions between POMs and H_2O are weak or only a small amount of H_2O can form hydrogen bonding with POMs. These findings based on ^1H NMR characterization are in good agreement with the explanation that most of the water molecules exist as H_2O clusters rather than $\text{DMF}\cdot n\text{H}_2\text{O}$ complexes in cases of high H_2O content ($V_{\text{H}_2\text{O}} > 20\%$). These results explain why the assembly of ordered structures cannot be achieved in systems with high H_2O content, that is, only water molecules

which exist as $\text{DMF} \cdot n\text{H}_2\text{O}$ complexes can precisely allocate POMs at the PEO region.

The above mentioned ^1H NMR characterization results are direct evidence to support our proposed SPEA mechanism. In the revised manuscript, we supplemented the ^1H NMR spectra results (see Fig. S8-10) and include the related description and discussion: ^1H nuclear magnetic resonance spectra suggest that PEO and POMs simultaneously interact with H_2O through hydrogen bonding in the DMF/ H_2O binary solvent system with appropriate compositions, implying that POMs interact with PEO-*b*-PS through the mediation of H_2O or $\text{DMF} \cdot n\text{H}_2\text{O}$ complexes (Supplementary Fig. 8-10)^{34,39}. (see Page 8). Detailed analysis is supplemented in the revised Supplementary Information (see Pages S19-21).

Figure R1. Schematic illustration of the underlying intermolecular interactions among PEO, non-acidic POMs and $\text{DMF} \cdot n\text{H}_2\text{O}$ complexes for the SPEA process

Figure R2. ^1H NMR spectra of PEO/POMs mixture recorded in DMF- d_7 (a, inevitably containing trace amounts of H_2O due to the hygroscopicity of DMF), DMF- $d_7/\text{D}_2\text{O}$ (b, 9:1 v/v) and DMF- $d_7/\text{D}_2\text{O}$ (c, 2:1 v/v) solution. The amount of solvent is fixed at 0.5 ml.

Figure R3. ^1H NMR spectra of POMs recorded in DMF- $d_7/\text{D}_2\text{O}$ (9:1 v/v) solution. The amount of solvent is 0.5 ml.

Figure R4. ^1H - ^1H NOESY spectrum of PEO/POMs/ H_2O /DMF mixture in CD_3Cl .

Reviewers' Comments:

Reviewer #4:

Remarks to the Author:

The authors have presented significant new NMR data that are sufficient to suggest the SPEA mechanism claimed in the article. Specifically the 2D NMR data confirm the association of water with PEO and are consistent with POM association with said water. Despite the lack of direct POM signal, this evidence is sufficiently suggestive to support the SPEA notion.